# RNA-binding proteins Musashi and tau soluble aggregates initiate nuclear dysfunction

Mauro Montalbano [1,2], Salome McAllen [1,2], Nicha Puangmalai [1,2], Urmi Sengupta [1,2], Nemil Bhatt [1,2], Omar D. Johnson [3], Michael G. Kharas [4] & Rakez Kayed [1,2 ✉]

Oligomeric assemblies of tau and the RNA-binding proteins (RBPs) Musashi (MSI) are reported in Alzheimer's disease (AD). However, the role of MSI and tau interaction in their aggregation process and its effects are nor clearly known in neurodegenerative diseases. Here, we investigated the expression and cellular localization of MSI1 and MSI2 in the brains tissues of Alzheimer's disease (AD), amyotrophic lateral sclerosis (ALS) and frontotemporal dementia (FTD) as well as in the wild-type mice and tau knock-out and P301L tau mouse models. We observed that formation of pathologically relevant protein inclusions was driven by the aberrant interactions between MSI and tau in the nuclei associated with age-dependent extracellular depositions of tau/MSI complexes. Furthermore, tau and MSI interactions induced impairment of nuclear/cytoplasm transport, chromatin remodeling and nuclear lamina formation. Our findings provide mechanistic insight for pathological accumulation of MSI/tau aggregates providing a potential basis for therapeutic interventions in neurodegenerative proteinopathies.

[1] Mitchell Center for Neurodegenerative Diseases, University of Texas Medical Branch, Galveston, TX 77555, USA. [2] Departments of Neurology, Neuroscience and Cell Biology, University of Texas Medical Branch, Galveston, TX 77555, USA. [3] School of Medicine, University of Texas Medical Branch, Galveston, TX 77555, USA. [4] Division of Molecular Pharmacology, Memorial Sloan Kettering Institute Cancer Center, New York City, NY, USA. ✉email: rakayed@utmb.edu

RNA-binding proteins (RBPs) facilitate ribonucleoprotein complex (RNP) formation to coordinate RNA processing and post-transcriptional gene regulation[1]. All RNA molecules are bound by specific sets of RBPs that regulate correct processing, transport, stability, and final degradation[2]. RBPs play roles in several RNA processing steps including transcription (TAF15, EWS, FUS), alternative splicing (TDP-43, FUS, TAF15, hnRNPA1, hnRNPA2/B1, EWS), alternative polyadenylation, transcript localization, sequestration into inclusions and stress granules, translation, RNA degradation, and turnover (MATR3)[3]. RBPs and impaired RNA processing have been implicated in the onset and progression of several neurodegenerative diseases[4]. Many cellular mechanisms are altered in neurodegenerative diseases, including the mislocalization of RBPs[5]. RBPs interact with tau to alter the physiology of stress granules and they aggregate in tauopathies[6]. Some notable RBPs such as Transactive Response DNA Binding Protein (TDP-43), Fused in Sarcoma (FUS), T-cell Intracytoplasmic Antigen (TIA-1)[7,8] and U1-70K[9] are dysregulated in neurodegenerative diseases. In our previous study, we have shown that the Musashi (MSI) family of RBPs is also involved in a cytotoxic interaction with tau in AD brains[10].

The MSI family comprises two homologs: Musashi1 (MSI1) and Musashi2 (MSI2). Each MSI protein homolog has two N-terminal RNA recognition motifs (RRMs)[11]. MSI1 presents with RRM1 (20-110) and RRM2 (109-186), as well as with Poly-Ala between 274-281aa. MSI2 has two RRM (21-110aa and 110-187aa). Furthermore, similar to MSI1, MSI2 presents with Poly-Ala 253-260aa (http://uniprot.org) (Fig. 1a).

MSIs are an evolutionarily conserved RBP family originally discovered in the central nervous system (CNS) and preferentially expressed in neuronal stem cells (NSCs)[12]. Mammalian MSI1 is expressed in fetal and adult NSCs and mature neurons[10]. MSI2's CNS expression pattern is similar to MSI1 in terms of high expression levels in neural stem/progenitor cells, and they have been postulated to play mutually overlapping roles that remain to be elucidated. Nevertheless, MSI2 is continuously expressed in a subset of CNS neurons, particularly GABAergic neurons[13].

We recently reported the first evidence of MSI interaction with tau[5,10]. Despite ectopic MSI1 in AD and Pick's Disease being reported many years ago[14], nothing has been found out about its pathogenesis until now. Some disease-associated RBPs also form pathological aggregates[15]. For example, mutations in RBPs can cause frontotemporal dementia (FTD), amyotrophic lateral sclerosis (ALS), spinocerebellar ataxia and myopathies[16–18]. Thus, RBP dysfunction appears to be involved in neurodegenerative pathogenesis.

We also reported that oligomeric MSI1 and MSI2 in human AD cortical tissue are found in the cytoplasm and nuclei of mature neurons and associate with oligomeric tau species[10]. Moreover, MSI proteins can form oligomers in vitro[10]. Clarifying MSI and tau cell compartmentalization and regulation are important steps toward identifying the crucial roles that they both play in the pathogenesis of AD and other forms of proteinopathies.

Tauopathies are characterized by accumulation of pathological tau in the somato-dendritic compartment and subsequent oligomer and fibril formation[10]. Growing evidence indicates the oligomeric forms are the toxic tau species involved in neuronal dysfunction and death[19,20]. However, there are other "non-canonical" intracellular tau locations. For, example non-phosphorylated tau localizes in the plasma membranes of different cell lines[21,22], as well as in lipid rafts in AD brain[23], mouse brain[24], and primary neurons[25]. Pathogenic phospho-tau was recently shown to disrupt nuclear-cytoplasmic transport in AD via binding the nuclear pore complex (NPC), e.g. to Nup98[26,27]. Collectively, these findings indicate that tau-associated proteins could shift the localization of tau and alter its functions, triggering neurodegeneration and toxic tau aggregation in various organelles including the nuclei. Intranuclear tau has been observed in wild-type mouse brain neurons[28,29] and both AD and control human brain[30]. However, we still do not fully understand the function of nuclear tau. It is necessary to clarify whether nuclear tau interacts with and influences DNA and determine its neurotoxic role in neurodegenerative pathologies[31,32].

In this study, we describe how MSI proteins accumulate and interact with tau oligomers in AD, ALS, and FTD. Comparing MSI protein accumulation between different neurodegenerative diseases allows for the possible identification of distinctive and/or common pathogenic mechanisms between AD, ALS, and FTD. By using an inducible tau HEK cell system (iHEK), nuclear perturbations were found. Furthermore, the cortical and hippocampal expression of MSI1 and MSI2 was recorded in C57, KO tau, and P301L tau mouse models at varying ages to identify any time-dependent expression.

Tau and MSI co-localize in the cytoplasm and nuclei of mouse neurons. Here we examined the effects of tau on MSI expression, localization and in vitro function. In particular, we investigated the effect of WT and mutant (P301L form) tau on MSI using iHEK cells that express WT and P301L mutant human tau forms and performed mass spectrometry analysis to study the interactome of toxic tau conformers and MSI1 in the presence of high cytoplasmic and nuclear levels of tau. We showed that tau modulates MSI protein levels and aggregation by regulating MSI's cellular localization and its interactome. This study is the first to show interaction between tau and MSI RBPs in multiple neurodegenerative diseases and suggest that pathophysiologies coupled by a nucleotoxic effect driven by tau through MSI dysfunction.

## Results

**MSI and tau form aggregates both in vitro and in human brains.** In our previous study, we showed that MSI1 and MSI2 can form their own aggregates in AD brains. We also observed oligomerization in vitro using human-recombinant MSI1 and MSI2[10]. Here, we mixed in vitro using the same oligomerization assay human-recombinant MSI and recombinant human-tau (2N4R) following established protocol[10] and briefly summarized in Fig. 1b. After co-incubation, the mixtures (MSI1 + TAU and MSI2 + TAU) were visualized with AFM. We observed the formation of oligomers from MSI1 and MSI2 (Fig. 1c, e, respectively), with the formation of large aggregates in the MSI1 mixture. We also co-incubated the two MSI proteins with tau monomer (2N4R), leading to the formation of large aggregates with evident changes in their shapes (Fig. 1d, f). Tau oligomers were prepared using an established protocol and were visualized with AFM (Supplementary Fig. 1a). The density of oligomers in the mixtures increased and their conformations changed drastically in both MSI1/TAU and MSI2/TAU mixtures. We also observed that MSI2 can form oligomers (Fig. 1e), but they are smaller in comparison to MSI1 oligomers (Fig. 1c). Surprisingly, mixing MSI2 and tau monomers led to the formation of large aggregates that were comparable with the aggregates observed in the MSI1/TAU mixture, indicating a high oligomerization property of MSI2 in the presence of tau (Fig. 1f).

To evaluate the different levels and distributions of MSI1 and MSI2 amongst varying neurodegenerative diseases, we co-stained AD, ALS, FTD, aged Control brains MSI1 (Fig. 1g) and MSI2 (Fig. 1h) with anti-tau oligomeric monoclonal antibody (TOMA-2). As expected, we observed high levels of TOMA-2 staining in diseased brains. Furthermore, in diseased brains, there was also a stronger signal coming from the MSI1 and MSI2 RBPs in comparison to the controls.

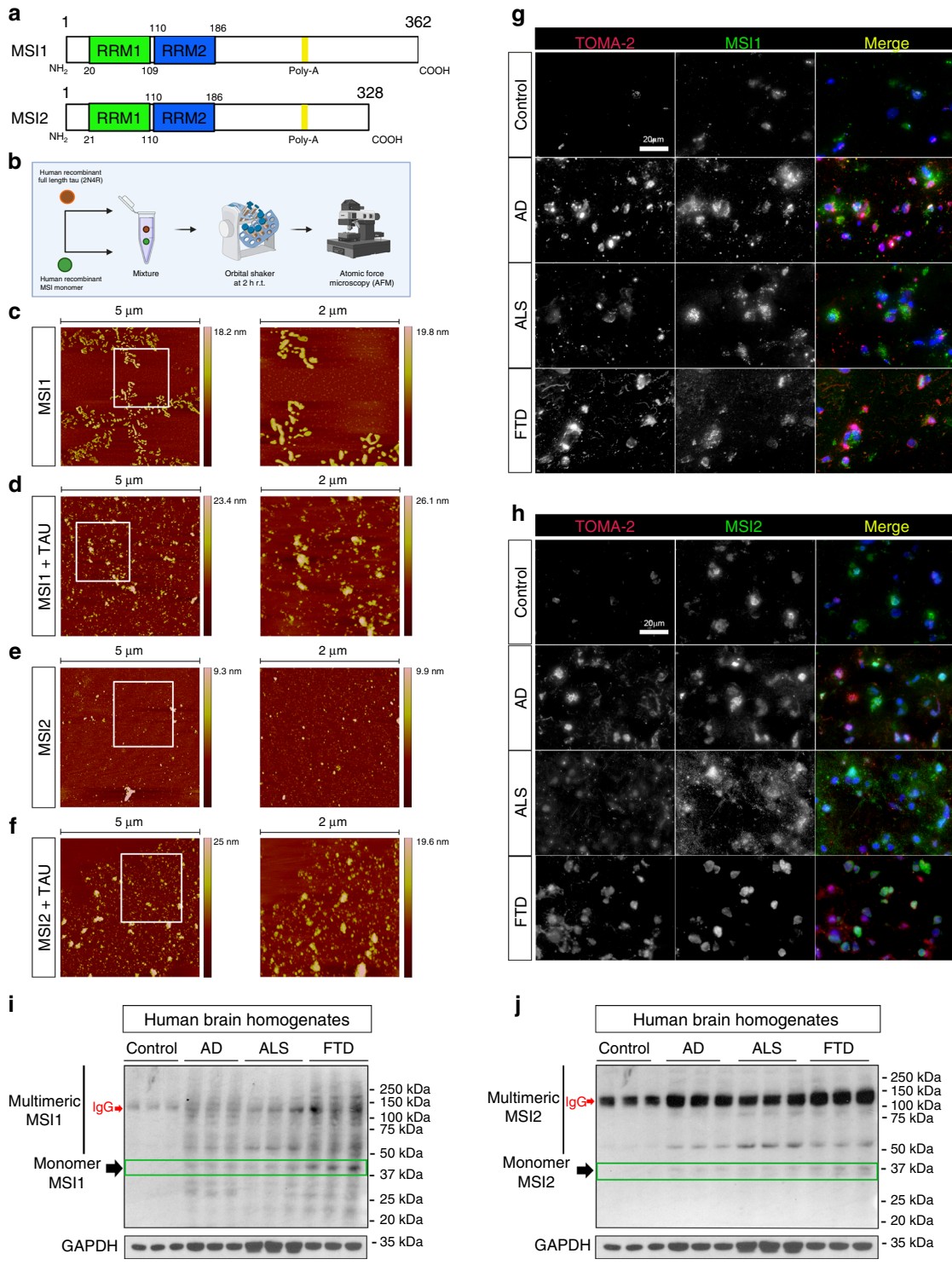

**Fig. 1 MSI/tau aggregation in vitro and MSI proteins in AD, ALS, and FTD frontal cortex. a** Schematic representation of MSI1 and MSI2 proteins (RRM; RNA Recognition Motif). **b** Diagram of experimental procedure to produce in vitro MSI and tau oligomers. **c** Representative 5 μm and 2 μm (white square in 5 μm image) AFM fields of MSI1 oligomers. **d** Representative 5 μm and 2 μm (white square in 5 μm image) AFM fields of MSI1/tau mixed oligomers. **e** Representative 5 μm and 2 μm (white square in 5 μm image) AFM fields of MSI2 oligomers. **f** Representative 5 μm and 2 μm (white square in 5 μm image) AFM fields of MSI2/tau mix oligomers. **g** Representative confocal images of Control, AD, ALS and FTD human brains. They were stained with TOMA-2 and MSI1 (showed in gray) and merge are presented in RGB (TOMA-2 red, MSI1/MSI2 green and DAPI blue; white scale bar: 20 μm). **h** Representative confocal images of Control, AD, ALS, and FTD human brains. Tissues were stained with TOMA-2 and MSI2 (shown in gray) and the merged channel is presented (TOMA-2-red, MSI2-green, DAPI-blue; white scale bar: 20 μm). **i–j** Western blot of MSI1 and MSI2 in control, AD, ALS, and FTD brain homogenates. GAPDH is also presented, and each sample is in triplicate (HMW: High Molecular Weight).

**Table 1 Brain tissues analyzed in this study from diseased and age-matched control subjects.**

| Clinical diagnosis | Case number | Age | Gender | PMI (h) | Braak stage (0-6) | Application |
|---|---|---|---|---|---|---|
| ALS | 2489 | 64 | M | 17 | 1 | IF/WB |
| ALS | 2498 | 68 | M | 24 | 1 | IF/WB |
| ALS | 1661 | 84 | M | 30 | 0 | Filter trap |
| ALS | 1779 | 65 | F | 9 | N/A | IF/WB |
| ALS | 1942 | 73 | F | 14 | N/A | WB |
| ALS | 1589 | 76 | F | 9 | N/A | WB |
| ALS | 1934 | 66 | F | 23 | N/A | WB |
| FTD | 1836 | 56 | F | 21 | 0 | IF/WB |
| FTD | 1841 | 60 | M | 7 | 0 | IF/WB/Filter trap |
| FTD | 5766 | 81 | M | 14 | | IF/WB |
| FTD | 5827 | 80 | F | 7 | 0 | WB |
| FTD | 5853 | 71 | M | N/A | N/A | WB |
| FTD | 5806 | 57 | F | 10 | N/A | WB |
| AD | 1120 | 83 | F | 4.75 | 6 | IF |
| AD | 1154 | 86 | M | 3.25 | 6 | IF |
| AD | 1098 | 81 | F | 2.75 | 5 | IF/WB/Filter trap |
| AD | 1964 | 67 | F | 33 | 6 | WB |
| AD | 5773 | 74 | M | 10 | 5 | WB |
| AD | 5779 | 73 | M | 15 | 6 | WB |
| AD | 5781 | 88 | M | 10 | 5 | WB |
| AD | 5829 | 68 | M | 12 | 6 | WB |
| Control | 5263 | 88 | M | 12.17 | 1 | IF |
| Control | 1161 | 84 | F | 2.50 | 0 | IF |
| Control | 1106 | 79 | M | 1.75 | 2 | IF/WB/Filter trap |
| Control | 1796 | 81 | M | 8 | 0 | WB |
| Control | 25-01 | N/A | N/A | N/A | 2 | WB |
| Control | 19-01 | N/A | N/A | N/A | 2 | WB |
| Control | 2-99 | 74 | F | 2.8 | 2 | WB |
| Control | 13-01 | 95 | M | 3.7 | 1 | WB |

*WB* western blot, *IF* iImmunofluorescence

To confirm the presence of MSI high molecular weights (HMWs) in diseased brains, we performed a Western blot from AD, ALS, FTD and control brain homogenates for MSI1 (Fig. 1i) and MSI2 (Fig. 1j). We observed an increment of MSI1 monomeric forms, but the most relevant difference was detected for HMWs that were particularly abundant in the MSI1 Western blot. For MSI2, we mainly observed the monomeric form and two major bands, the first one being approximately 50 kDa and the second one being approximately 125 kDa. Large tissue IF staining fields (×10 magnification, Supplementary Fig. 2a, b) and WB (Supplementary Fig. 2c, d) from all diseased brains of MSI1 and MSI2 were shown respectively, and the relative quantification of MSI monomeric and multimeric forms is represented in Supplementary Fig. 2e–h. A11/Tau13 immunoblots and secondary antibodies alone membranes are represented in Supplementary Fig. 1b, c, respectively. Validation of the specific anti-MSI1 band(s) by siRNA experiment is shown by MSI1 immunoblot and relative quantification (Supplementary Fig. 1d–f). Li-Cor immunoblot is also included for A11 and Tau13 antibodies in Supplementary Fig. 1g. Validation of MSI1 and MSI2 antibodies has shown in total lysate of iHEK cell model and human brains represented in the same blot (Supplementary Fig. 1h, i). The information from all the patients used for this study are summarized in Table 1.

**Musashi present intra-subject variability in AD, ALS, and FTD.** In Fig. 1g–h, we noticed high distributive variability for MSI1, MSI2, and TOMA-2 inside the same samples. These observations are summarized in a schematic experimental plan to characterize MSI and tau proteins in different diseases (Fig. 2a). We mainly observed in AD spotted nuclear and perinuclear MSI1, while there was also large TOMA-2 aggregates in the nuclei and cytoplasm of neurons (Fig. 2c). Additional variability was detected in FTD brain cells where MSI1 presented different patterns (Fig. 2d). TOMA-2 also presented a large staining pattern type from perinuclear to compact signals, as well as a spotted distribution. Similar observations were made for ALS brains, except that the patterns and intensities of TOMA-2 were found to be less pronounced as those in AD and FTD (Fig. 2e). MSI1/TOMA2 staining of control (Ctr) tissue is reported (Fig. 2b). The common element between the three different diseases has been evaluated and quantified in the MSI/Tau PCC (Pearson Correlation Coefficient) value, expressing co-localization measurements. It was significantly higher among the discussed neurodegenerative diseases than in the control (Fig. 2f). These observations indicate that the interactions and colocalization of MSI proteins with tau oligomers are involved in AD, ALS and FTD at variable degrees, which probably exert different effects on the cytoplasm and nuclei of neurons. We compared by slot plot MSI1 and MSI2 in soluble and insoluble fractions from all the brains, we observed large amount of MSI1 and MSI2 in AD soluble fractions compared to other samples. Furthermore, we also detect them in ALS and FTD with a weak signal from control (Fig. 2g- Soluble Fraction). MSI1 and MSI2 are present in insoluble fractions of AD brains (Fig. 2g —insoluble fraction), low signal is detected from Ctr, ALS and FTD. Tau13 slot plot has been performed to confirm tau accumulation in disease brains (Fig. 2h). Secondary (Rabbit and Mouse) antibody alone immunoblot is presented in Supplementary Fig. 3a, b, respectively). Tau13 immunoblot (IB) also is presented (Supplementary Fig. 3c). We also performed immunostaining in the same diseased brains for MSI2 and TOMA-2 (Supplementary Fig. 3d–g).

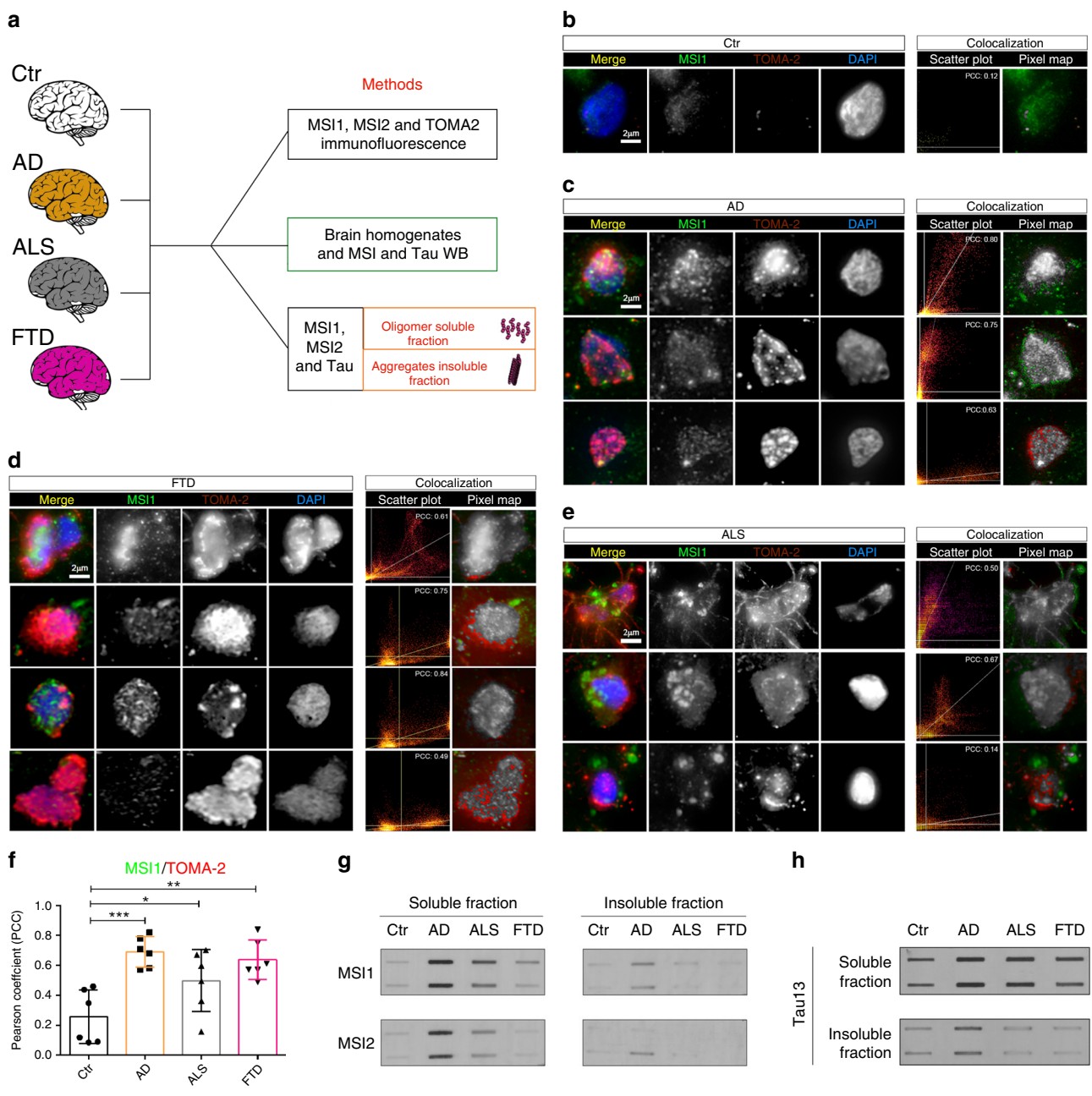

**Fig. 2 Intra-subject variability of Musashi deposition in AD, ALS, and FTD frontal cortex. a** Schematic approach to study MSI1, MSI2 and Tau deposition in human diseased brains. **b** Representative confocal image of nuclei stained with MSI1, TOMA-2 and DAPI in control (Ctr) brains. **c** Confocal images of cortical AD brain showed nuclei stained with MSI1, TOMA-2, and DAPI, three different nuclei are represented for each field the pixel distribution graph and colocalization pixel map has been reported. **d** Confocal images of cortical FTD brain showed nuclei stained with MSI1, TOMA-2 and DAPI, three different nuclei are represented for each field the pixel distribution graph and colocalization pixel map has been reported. **e** Confocal images of cortical ALS brain showed nuclei stained with MSI1, TOMA-2 and DAPI, three different nuclei are represented for each field the pixel distribution graph and colocalization pixel map has been reported. In all images merge is presented with MSI1 (green), TOMA-2 (red) and DAPI (blue) white scale bar: 2 μm. **f** PCC comparison between MSI1 and TOMA-2 intensities, significant co-localization has been observed between control and diseased brains. In particular, Ctr vs. AD ***, Ctr vs. ALS *, Ctr vs FTD **, Dunnett's multiple comparisons test One-way ANOVA has been performed with a *p*-value=0.0006 between columns. $N = 3$ biologically independent cells (ROIs) examined over three independent experiments. Data are presented as mean ± SD. **g** MSI1 and MSI2 soluble and insoluble fraction from Ctr, AD, ALS, and FTD human brains, presented in duplicate. **h** Tau soluble and insoluble fractions from Ctr, AD, ALS, FTD human brains, presented in duplicate.

**MSI1 accumulates in the nuclei and axons of aged P301L mice.** The distribution of MSI protein in different mouse models (C57BL/6, Tau KO and P301L) was studied over time to determine whether or not the MSI distribution was associated with aging. In particular, we stained for MSI1 and MSI2 brain slides from C57BL/6, Tau KO and Tau P301L models. We particularly focused our attention on the P301L model, because our previous cell data showed an abnormal accumulation of MSI proteins in the presence of higher levels of tau[5]. In 2-month-old animals, nuclear MSI1 localization in the dentate

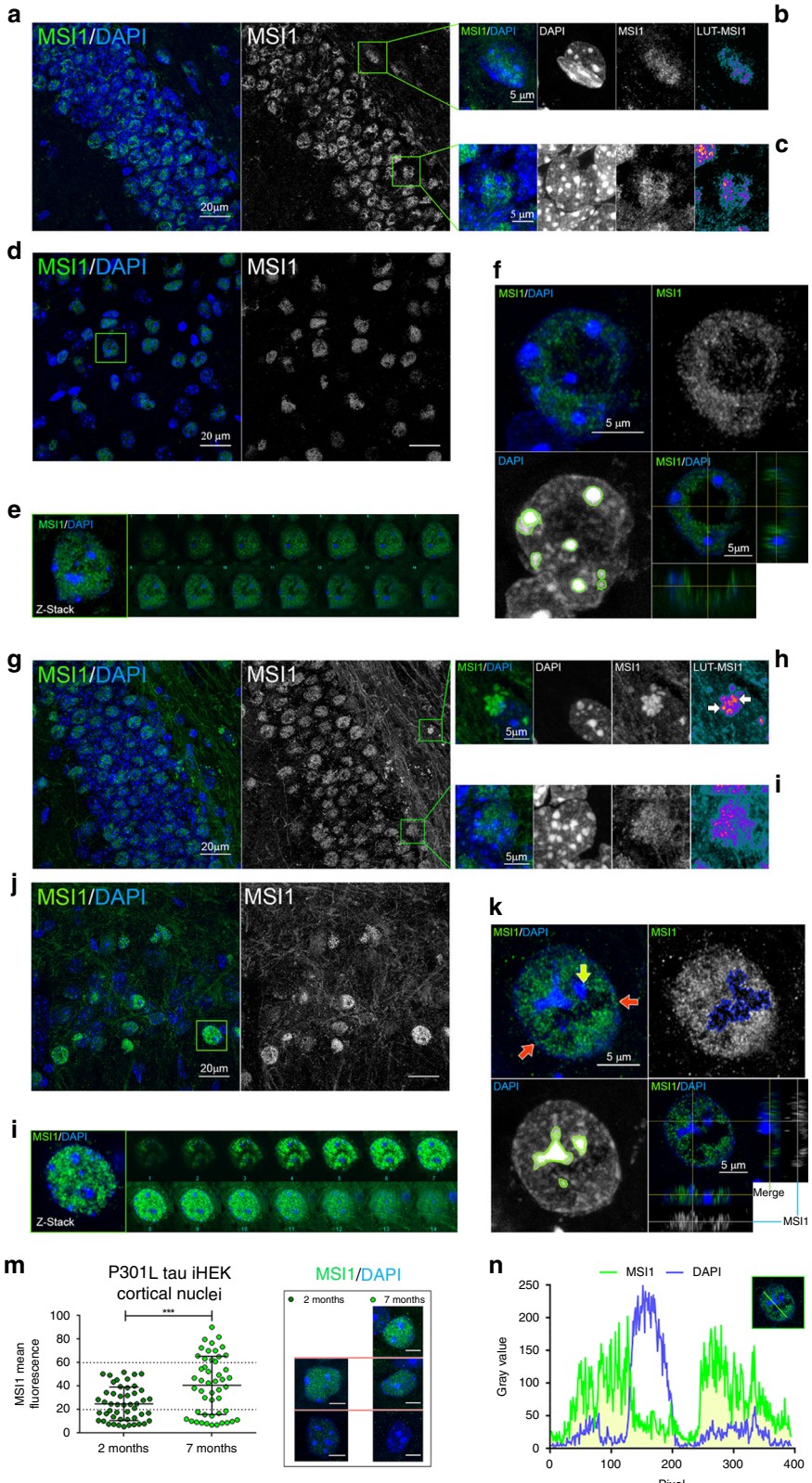

gyrus (DG) (Fig. 3a–c) and in the cortex (CTX) (Fig. 3d) were observed. There was also a spot distribution of MSI1 in the nuclear cortex (Fig. 3e), which was confirmed in the orthogonal view (Fig. 3f). In aged, 7-month-old mouse brains, there were pronounced changes in the distribution and level of MSI1 in both the DG (Fig. 3g–i) and the cortex (Fig. 3j). We observed extracellular deposition of dense MSI1 in the hilus (Fig. 3h,

white arrows) and an evident positive signal in the Mossy fiber. Our finding suggested that in the cortex, there was an elevated MSI1 level nuclei population compared to the young brain, implying that there was indeed an age driven factor to MSI distribution. In particular, we observed a high-density MSI1 nuclei (Fig. 3k), MSI1 was observed in the nucleoli periphery (yellow arrow) and in the nuclear envelop (red arrows), this

**Fig. 3 MSI1 accumulate in the nuclei and axons of aged P301L mice. a** Representative confocal image of 2-month-old P301L DG (MSI1—green, DAPI—blue), MSI1 channel is presented in gray, white scale bar: 20 μm. **b** Inset of subgranular zone (SGZ) nucleus enlarged from (**a**), merge MSI1. DAPI and Look-Up Table Fire (LUT)-MSI1 channels are represented, white scale bar: 5 μm. **c** Inset of DG nucleus enlarged from A. Merge, MSI1, DAPI and LUT-MSI1 channels are represented, white scale bar: 5 μm. **d** Representative confocal image of 2-month-old P301L cortex (MSI1—green, DAPI—blue), MSI1 channel is presented in gray, white scale bar: 20 μm. **e** Cortical nucleus (highlighted in D) and montage of Z-stack. **f** Merge (MSI1-green and DAPI—blue), MSI1 (gray) and DAPI channels are presented for cortical nuclei with orthogonal view (bottom right), white scale bar: 5 μm. **g** Representative confocal image of 7-month-old P301L DG (MSI1—green, DAPI—blue), MSI1 channel is presented in gray, white scale bar: 20 μm. **h** Inset of SGZ nuclei enlarged from (**g**), merge MSI1. DAPI and LUT-MSI1 channels are represented, white scale bar: 5 μm. **i** Inset of DG nucleus enlarged from (**g**). Merge, MSI1, DAPI, and LUT-MSI1 channels are represented, white scale bar: 5 μm. **j** Representative confocal image of 7-month-old P301L cortex (MSI1—green, DAPI—blue), MSI1 channel is presented in gray, white scale bar: 20 μm. **k** Merge (MSI1—green and DAPI—blue), MSI1 (gray) and DAPI channels are presented for 7-month cortical nuclei with orthogonal view (bottom right), white scale bar: 5 μm. **l** 7-month cortical nucleus (highlighted in (**j**)) and montage of Z-stack. **m** Quantification of cortical MSI1 mean fluorescence in 2- and 7-month-old P301L mice nuclei, significant increment of MSI1 has been observed (2 M vs 7 M $p = 0.0002$*** unpaired $t$-test), representative images from each group are shown (white scale bar: 5 μm). $N = 50$ biologically independent nuclei (ROIs) examined over three independent experiments. Data are presented as mean ± SD. **n** Nuclear MSI1 and DAPI fluorescence profiles revealed a not homogenous distribution of MSI1 in the nuclei and no overlap with nucleoli (white line profile: 10 μm).

more dense and compact MSI1 in the nuclei is shown in the z-stack and montage in Fig. 3l. We quantified and compared the nuclei population for MSI1 intensity at 7-month in the cortex and observed a strong signal (Fig. 3m). We also observed in cortical neuronal nuclei that MSI1 is not associated with the nucleoli and that some nucleoplasmic area was not occupied by the protein (Fig. 3n). Collectively, these observations suggest that MSI1 accumulates in the nuclei in aged mice and that it is not homogenously distributed in the nucleoplasm. Changes in MSI forms and amount in all three models were observed, and displayed in Fig. 4.

**Age-dependent MSI distribution in P301L mice.** In general, we detected in P301L 2-month-old mice a nuclear and axonal MSI1 distribution in hippocampi and cortex (Fig. 4a). In 7-month-old mice, we observed a prominent expression at the nuclear level of MSI1 in the hippocampi and cortex (as observed in Fig. 3) even in the axons using a confocal microscope (Fig. 3g). An evident difference between the two ages was the presence of nuclear and perinuclear aggregate-like structure mainly present in CA3 but detected also in DG, CA1, and CTX (Fig. 4b). MSI2 showed in 2-month-old mice hippocampi a low expression in all the area analyzed (Hippocampi and Cortices, Fig. 4c), more signal has been observed in axons. Interestingly, we observed numerous MSI2 positive cells in Subgranular zone (SGZ) of DG in 7-month-old mice compared to 2-month (Fig. 4d). SGZ of hippocampus showed numerous MSI2 positive cells, their position and shape suggest them to be neuronal progenitor stem cells. Other MSI2 positive cells are detected in area (CA1, CA3, and CTX) not typically involved in neurogenesis process. MSI2 looks in these cells cytoplasmic with lower signal in the nuclei. Age-dependent changes have also been observed in 2- and 7-month-old C57 (Supplementary Fig. 4) and Tau KO (Supplementary Fig. 5) mice. In 7-month WT mice, we observed extensive MSI1 immunostaining in the cytoplasm of hippocampus and cortex and abundant cytoplasmic MSI2 labeling in the CA1 layer that was less prominent presence in the CA3 and DG of the hippocampus. In Tau KO mice, there was significant age-related increase in MSI1 level (and change in MSI2 distribution). These observations indicate that the presence/absence of tau but also its form (WT/P301L tau) modulate the localization and abundance of MSI proteins in the hippocampi and cortices of mouse models. Interestingly, MSI proteins showed an age-dependent distribution in all mouse models studied. This aspect provides insights for further study that could elucidate the direct effect of tau on MSI expression and localization during aging processes in mice.

**MSI/tau complexes localize in P301L neuronal nuclei.** To study in situ interactions of MSI and tau oligomers in mouse brains, we performed a PLA[33] using Tau13, TOMA-2, MSI1, and MSI2 antibodies to detect protein complexes in the hippocampus and cortex of P301L mice (Fig. 5). PLA in P301L tau cortex is presented in Supplementary Fig. 6a and MSI1/TOMA-2 speckles area distribution in the hippocampus is represented in Supplementary Fig. 6b, c. We performed in parallel co-immunofluorescence using the same primary (MSI1, Tau13, and TOMA-2 Supplementary Fig. 7) antibodies to compare with the PLA signal. MSI1 in 2-month-old P301L was observed at lower levels in the nuclei of DG and CA layers. Axonal signals that were in the hilus and stratum lacunosum moleculare were primarily detected. In 7-month-old P301L tested with MSI1/TOMA-2, the PLA signal showed an impressive accumulation of MSI1 in the nuclei of neurons when compared to younger ones (Fig. 5a, right panel). In animals of equal age, using MSI2 and TOMA-2 antibodies, we observed a lower level of MSI/tau interactions in 2-month-old mice with evident increments present the hippocampus and cortex from the PLA signal (Fig. 5b). Imaging at higher magnifications allows us to appreciate the Tau13/MSI1 signal, which was characterized by extracellular speckles, present in CTX, DG, CA1, and CA3 (Fig. 5c–f, upper panels).

The same brains were also stained for MSI1/TOMA-2 combinations. Interestingly, there was an extracellular speckle pattern comparable with Tau13, but a nuclear diffuse signal was absent in the MSI1/Tau13 PLA slides. This behavior has been detected in CTX, DG, CA1 and CA3 (Fig. 5c–f, lower panels). The difference revealed by PLA showed a peculiar immunoreactivity of nuclear tau oligomers by TOMA-2, confirmed in human AD brains orthogonal view (Supplementary Fig. 6d) and by canonical IF (Fig. 1g–h). We quantified the percentage of MSI1/TOMA-2 PLA speckles in four areas and observed that the majority was concentrated in the DG (34.66%), but significant amounts were also present in CA1 (24.43%), CA3 (23.52%) and CTX (17.38%) (Fig. 5k). Even if the majority of aggregates occurred in the DG, the largest aggregates were found to be in the CA3 regions (Fig. 5l).

MSI2 showed a unique localization pattern compared to MSI1. In 2-month-old mice, hippocampal nuclei appear to be MSI2 negative. Furthermore, axons that are mainly in the CA1 region appear to be positive for MSI2. In contrast to MSI1, we observed a different pattern of MSI2 in 7-month-old mice, particularly that several cells were positive for MSI2 in the subgranular zone (SGZ) of hippocampi, suggesting progenitor stem cell behavior (due to their morphologies and location). We also observed positive cells all around the hippocampus, including the CA3 and CA1 regions.

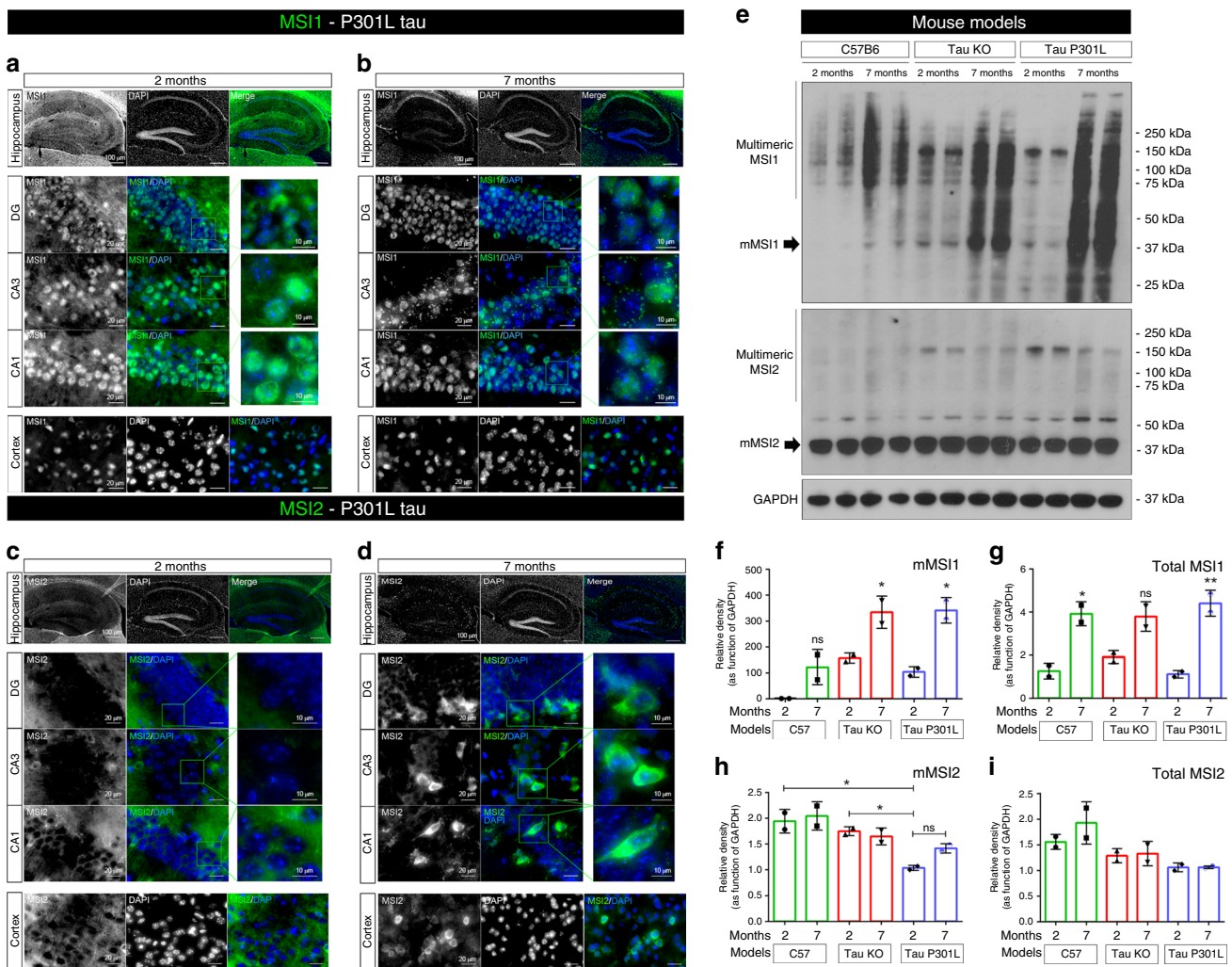

**Fig. 4 MSI proteins in hippocampus and cortex of P301L tau mouse model. a** Representative MSI1 (gray/green) immunofluorescence of 2-month P301L hippocampus (10X, white scale bar: 100 μm). High Magnification images (100X) of DG, CA3, CA1 and Cortex are presented (white scale bar: 20 μm). **b** Representative MSI1 (gray/green) immunofluorescence of 7-month P301L hippocampus (10X, white scale bar: 100 μm). High Magnification images (100X) of DG, CA3, CA1 and Cortex are presented (white scale bar: 20 μm). **c** Representative MSI2 gray/green) immunofluorescence of 2-month P301L hippocampus (10X, white scale bar: 100 μm). High Magnification images (100X) of DG, CA3, CA1, and Cortex are presented (white scale bar: 20 μm). **d** Representative MSI2 gray/green) immunofluorescence of 7-month P301L hippocampus (10X, white scale bar: 100 μm). High Magnification images (100X) of DG, CA3, CA1, and Cortex are presented (white scale bar: 20 μm). For all the images, MSI2 is presented in gray channel and in the merge MSI2 is presented (Nuclei are stained with DAPI, blue). All zoomed images of DG, CA3 and CA1 have a white scale bar of 10 μm (**e**) MSI1, MSI2, and GAPDH immunoblots in 2- and 7-month-old mouse brain homogenates (C57, tau KO, and tau P301L models). Immunoblot samples are presented in duplicate. **f–g** Relative density of monomeric and total MSI1, respectively is presented (Monomeric MSI1 One-way ANOVA, Tukey's multiple comparisons test has been performed (*p*-value=0.0014 between the columns, for total MSI1 *p* = 0.0014). Data are presented as mean ± SD. **h, i** Relative density of monomeric and total MSI2 respectively is presented. All relative density are in function of GAPDH. One-way ANOVA, Tukey's multiple comparisons test has been performed to compare MSI relative amounts among the ages and models (monomeric MSI2 *p* = 0.0084, for total MSI2 *p* = 0.0415). Sample size for data in (**f–i**) is *n* = 3 biologically independent samples examined over three independent experiments, two independent samples are shown in the WB. Data are presented as mean ± SD.

These observations indicate that MSI1 and MSI2 are expressed in different cellular populations and proliferate in an age-dependent manner. The two homologs also seem to be expressed by two different neuronal cell types. MSI1 is expressed in mature neurons of the hippocampus while MSI2 (7-month) is expressed in progenitor stem cells, even if low cytoplasmic levels have been detected in many mature neurons (Supplementary Fig. 4). These distributional differences between MSI1 and MSI2 indicate possible changes in function during the aging of the P301L mice, supporting the case for possible pathological missorting of MSI proteins with the formation of extracellular aggregates and cytoplasmic/nuclear accumulation (Fig. 5g–j).

Additionally, the cortical changes of these animals were observed. MSI1 showed more positive nuclei in 7-month-old mice, but maintained nuclear localization. MSI2 modified its localization from an axonal to a cytoplasmic distribution. As observed in the hippocampi, MSI proteins changed completely as their cellular localization also gradually appeared in the cortex with aging. Furthermore, extracellular aggregates of MSI2/TOMA-2, as well as those of MSI1, were present in 7-month-old mice (Fig. 5g–j, lower panels).

**P301L tau destabilizes nuclear morphology and transport.** We observed in a previous study that there was a strong reduction of

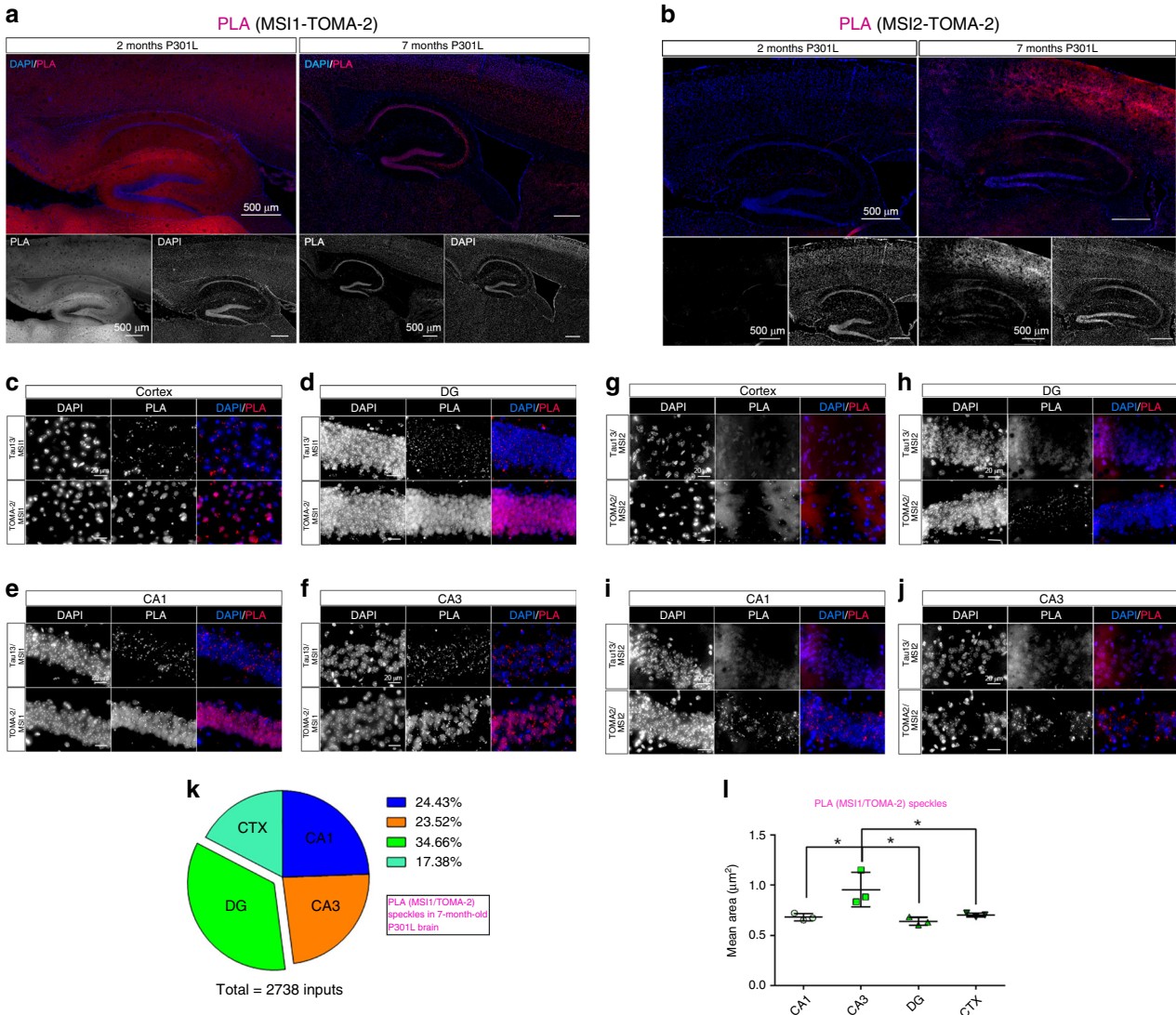

**Fig. 5 MSI/tau complexes localize in P301L hippocampus and cortex nuclei in aged mice. a** Representative hippocampus of 2- and 7-month-old P301L mice. PLA (MSI1-TOMA-2 combination) represented in red and DAPI (blue) stained nuclei. PLA and DAPI channels are represented in gray below the stitched panels. **b** Representative hippocampus of 2- and 7-month-old P301L mice. PLA (MSI2-TOMA-2 combination) represented in red and DAPI (blue) stained nuclei. PLA and DAPI channels are represented in gray below the stitched panels. 10x magnification and white scale bar: 500 μm. **c** Cortex 7-month PLA Tau13/MSI1 (top) and TOMA-2/MSI1 (bottom). **d** DG 7-month PLA Tau13/MSI1 (top) and TOMA-2/MSI1 (bottom). **e** CA1 7-month PLA Tau13/MSI1 (top) and TOMA-2/MSI1 (bottom). **f** CA3 7-month PLA Tau13/MSI1 (top) and TOMA-2/MSI1 (bottom). **g** Cortex 7-month PLA Tau13/MSI2 (top) and TOMA-2/MSI2 (bottom). **h** DG 7-month PLA Tau13/MSI2 (top) and TOMA-2/MSI2 (bottom). **i** CA1 7-month PLA Tau13/MSI2 (top) and TOMA-2/MSI2 (bottom). **j** CA3 7-month PLA Tau13/MSI2 (top) and TOMA-2/MSI2 (bottom). Magnification ×100 and white scale bar: 20 μm. **k** Percentage of MSI1/TOMA-2 foci in CTX, DG, CA1 and CA3 in brains from 7-month-old mice. Percentage is represented by a pie chart. **l** Mean Area quantification of MSI1/TOMA-2 foci in CTX, DG, CA1, and CA3 in brains from 7-month-old mice (one-way ANOVA, * $p = 0.0106$ between the columns). Three ROIs from $n = 3$ biologically independent examined over three independent experiments. Data are presented as mean ± SD.

LaminB1 and Histone3 levels in P301L iHEK in the presence of high MSI/tau levels, indicating nuclear discomfort[5]. We stained for LaminA/C P301L iHEK cells and observed rough and initial invagination (green arrows) of the nuclear membrane at 12 h after Tet induction (Fig. 6a, b). This invagination reached its maximum size at 24 h after Tet treatment (Fig. 6c, green arrows). Reduction of LaminA/C was confirmed via Western blotting of the nuclear fraction of cells (Fig. 6d), and increments of cell fractions with nuclear membrane invaginations was quantified and plotted, as shown in Fig. 6e. Mass spectrometry analysis confirmed a big reduction in the nuclear fraction of 24 h induced P301L in different nuclear membrane proteins: Emerin, LaminB receptor, Man1 and Nesprin 2 (Fig. 6f). Chromatin connectors

such as HP1 and BAF, were reduced (Fig. 6g), as well as Lamin-associated proteins 2 (LAP2: isoform α and isoform β/γ) (Fig. 6h). We also evaluated the distributions of nucleoporins (NUP) in cytoplasmic and nuclear compartments. There were not marked shifts in NUP levels in the nuclear fraction. Very few NUP were substantially decreased (Nup43, Nup153, Nup50, Nup214) and few of them even showed a slight increment in their level (Nup88, nuclear pore glycoprotein p62, Nup98). Nup133, and Nup155 showed a remarkable increment after Tet induction (Fig. 6i). Interestingly, a larger level of Nup was detected in the cytoplasmic fraction after induction, particularly in Nup153, Nup88, Nup50, Nup85, Nup107, Nup133, Nup214, and Nup155. We also found Nup after induction that were not represented in

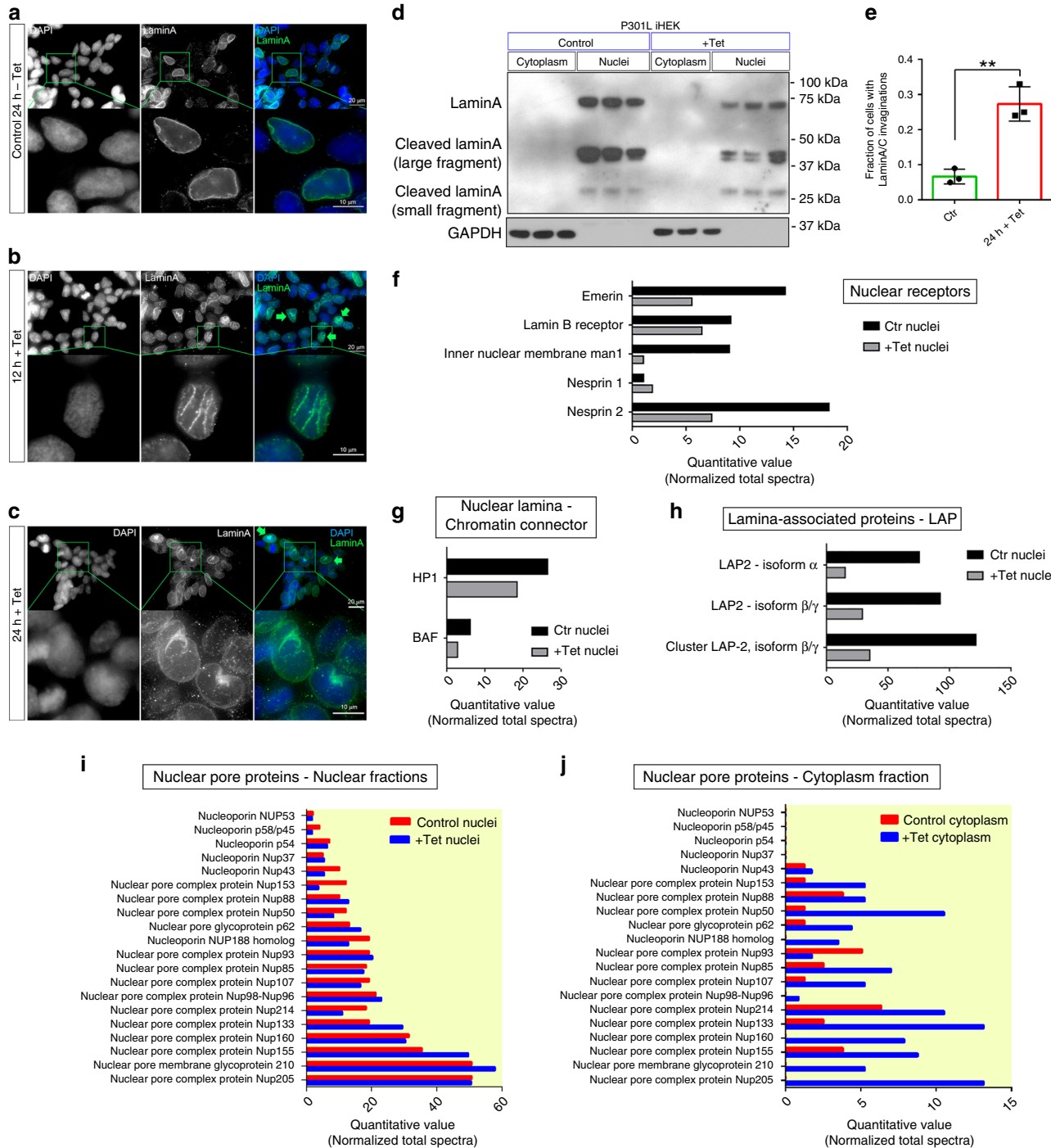

**Fig. 6 P301L tau destabilizes nuclear morphology and transport. a–c** Representative images of P301L tau iHEK at 0 h, 12 h, and 24 h after Tet induction, respectively. Cells are stained with LaminA (green) and DAPI (blue), magnification ×100 white scale bar: 20 µm, enlarged ROI white scale bar: 10 µm. **d** Western blot of LaminA in cell fractions (cytoplasm and nuclei) of P301L tau iHEK (Control and Tet). GAPDH has been used for cytoplasmic fractions purity. **e** Fraction of cells with LaminA invaginations, (two-tailed unpaired t-test **, p = 0.0026). N = 3 biologically independent (ROIs) examined over three independent experiments. Data are presented as mean ± SD. **f** MS quantitative value of nuclear receptor: Emerin, Lamin B receptor, Man1, Nesprin1, and Nesprin2. **g** MS quantitative value of nuclear lamina/chromatin connector: HP1 and BAF. **h** MS quantitative value of lamina-associated proteins. **i** MS quantitative value of nuclear fraction of NUPs. **j** MS quantitative value of cytoplasmic fraction of NUP.

the control, such as Nup188, Nup98, Nup160, Nup210, and Nup205 (Fig. 6j). We evaluated the formation of nuclear membrane invaginations after 6 h from Tet induction and found membrane bumping (Supplementary Fig. 8a). It was also evaluated whether the nuclear membrane defects were reversible when excluding Tet from the culture medium. In addition, after 24 h and 48 h (Tet OFF), we observed a recovery of nuclear

membrane bumping and integrity (Supplementary Fig. 8b). Reduction of LaminB1 was previously evaluated by Western blot[5] and confirmed by MS (Supplementary Fig. 8c). In addition, other nuclear components have been affected by high levels of tau. In particular, Histone levels were considerably reduced after Tet induction (Supplementary Fig. 8d), and Importins also appeared to be dysregulated as well (Supplementary Fig. 8e).

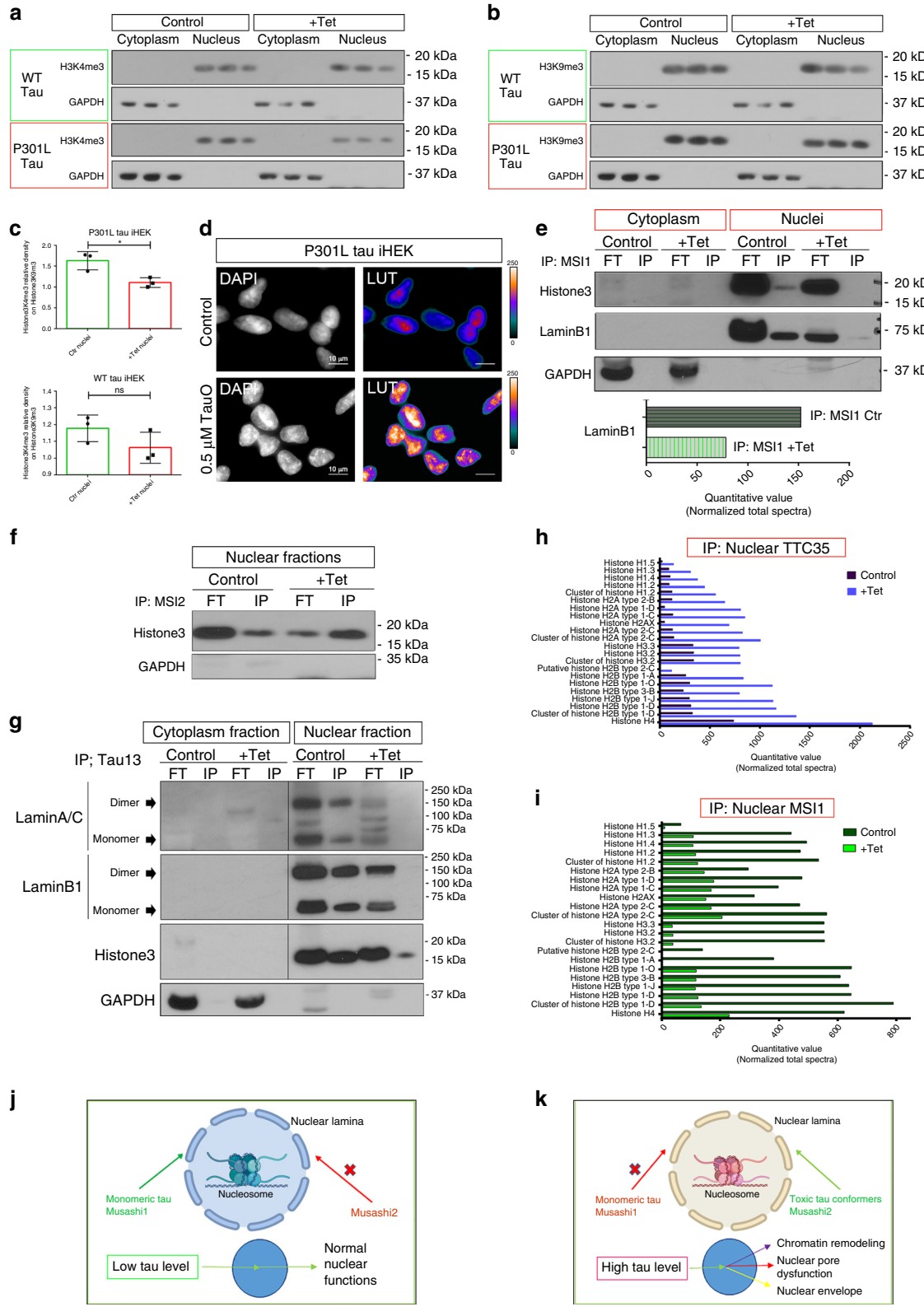

Tau increment in +Tet cells has been confirmed by MS (Supplementary Fig. 9d).

**MSI and tau bind Histone3 modulating chromatin state in the cells.** We evaluated markers of chromatin condensation by Western blot to establish if high levels of WT and P301L tau could modulate DNA packaging. As markers for chromatin state we used for euchromatin H3K4me3 and for heterochromatin H3K9me3. Western blot analysis (Fig. 7a, b) showed a significant decrement of H3K4me3 from in the nuclei of P301L tau iHEK after treatment with Tet. In WT tau iHEK, it showed a slight decrement, but it was not significant (Fig. 7c). H3K4me3 is a marker of activation of the chromatin; the concomitantly

**Fig. 7 MSI and tau bind Histone3 modulating chromatin state in the cells. a, b** H3K4me3 and H3K9me3 WB in cytoplasmic and nuclear fractions of WT and P301L tau iHEK. **c** Relative density of H3K4m3 in P301L (two-tailed unpaired *t*-test; Ctr vs Tet, * *p* = 0.0217) and WT iHEK (two-tailed unpaired *t*-test; *p* = 0.1758; ns). *N* = 3 biologically independent cells (ROIs) examined over three independent experiments. Data are presented as mean ± SD. **d** DAPI staining of nuclei control and TauO 0.5 µM treated P301L tau iHEK. Gray channel and LUT Fire are represented. White scale bar: 10 µm. **e** IB of Histone3, LaminB1, and GAPDH of IP MSI1 cytoplasm and nuclear fractions of control and Tet P301L tau iHEK. **f** IB of Histone3 and GAPDH of IP MSI2 nuclear fractions of control and Tet P310L tau iHEK. **g** IB of LaminA, LaminB1, Histone3, and GAPDH of IP Tau13 cytosolic and nuclear fractions of control and Tet P301L tau iHEK. All IB present Input (FT) fractions. **h** MS of IP TTC35 nuclear fraction of Histone proteins. **i** MS of IP MSI1 nuclear fraction of Histone proteins. **j, k** Normal and pathological, respectively, effects of mutant P301L tau oligomers in the cell model. Folded Monomeric tau and nuclear MSI1 regulate nuclear activity, while toxic tau conformers and nuclear MSI2 unbalanced nuclear homeostasis modulating chromatin remodeling, nucleoporin localization and destabilizing nuclear envelope.

unchanged amounts of H3K9me3 levels suggested that high level of P301L tau improved chromatin condensation in the nuclei. Comparison of DAPI density in P301L tau iHEK cells with/ without TauO (0.5 µM) was conducted to evaluate the different condensation states of the DNA (Fig. 7d), which confirmed an incremented condensation of chromatin within the nuclei in the presence of oligomeric tau. Montage and z-stacks of representative DAPI staining into the nucleus from Ctr and TauO treated cell has been reported in Supplementary Fig. 9f, g, respectively.

To establish which interactions occur between tau, Histone3 and/or Lamins, we performed immunoprecipitation (IP). Nuclear IP fractions of MSI1 and MSI2 (Fig. 7e, f) revealed association with Histone3 of both MSI proteins. Interestingly, in the MSI2 IP nuclear fraction, we observed an increment of binding between Histone3 and MSI2 after Tet induction while the MSI1 IP showed reduced interaction. This observation indicates a compensatory effect of MSI2 due to the lower association of MSI1 with Histone3. It might also suggest a gain of function (GOF) of MSI2 that, in the presence of high levels of tau, acquires more affinity to nucleosome components. Given these observations, it follows that there could be a potential synergistic relation between tau and MSI1 at the nucleosomal level, especially since MSI1 is altered in the presence of oligomeric tau (which exists in small quantities within the nucleus). High level of WT tau induce more interaction between MSI2 and Histone3, but at a different magnitude (Supplementary Fig. 9a).

In the Tau13 IP nuclear fraction, there was a dramatic reduction of LaminB1, LaminA, and Histone3 levels associated with tau after induction with Tet (Fig. 7g, right panel). In particular, interaction with LaminA and LaminB1 after Tet induction is completely lost. MS analysis of TTC35 IP nuclear fractions confirmed an incremented interaction of toxic tau conformers with Histones in Tet treated cells (Fig. 7h) with a concomitant reduction of association between MSI1 and Histones (Fig. 7e) confirmed also with WB. Furthermore, after Tet induction toxic tau species bind LaminB1, while in opposite MSI1 lost interaction with LaminB1 itself, as well as with Histone3 (Fig. 7e, i). Cytoplasmic FT/IP TTC35 fractions have been tested for purity incubating with Histone 3 (Supplementary Fig. 9c) and as expected not signal was detected. Also, Lamina-associated proteins (LAPs) in Tet induced P301L tau iHEK has been observed reduced in MSI1 IP nuclear fractions by MS (Supplementary Fig. S9e). MS confirmed the increment of tau levels (cytoplasm and nuclei) in P301L iHEK incubated with Tet (Supplementary Fig. 9d).

The results suggest that at low levels of tau, MSI1 and MSI2 bind to nucleosomes and nuclear lamina. This synergistic action keeps nuclear function unaltered (Fig. 7j). In contrast, high levels of tau produce toxic forms that bind to nucleosomes and lamina, improving MSI2 affinity for them and shooting down MSI1 interactions. This unbalanced pathway in the nuclei affects chromatin remodeling, nuclear pore and nuclear envelop dysfunctions, that all work together to trigger a stress response

in the cell (Fig. 7k). These results indicate a strong association of tau/chromatin, suggesting potential function on DNA packaging that is yet to be understood.

## Discussion

In this study, we reported the first evidence of intracellular and extracellular depositions of MSI1 and MSI2 in ALS and FTD, in addition to AD. These depositions were characterized in all three pathologies by a strong expression of MSI1 and MSI2 in cortical human tissues. Interestingly, as their protein levels increased, we noted stronger co-localization with tau oligomers compared to age-matched control brains. MSI depositions were present in the three pathologies, and each disease exhibited a different aggregation pattern.

Starting from in vitro experiments, there was a noteworthy effect between monomeric tau and MSI proteins, in that both tau and MSI joined to form large oligomeric aggregates. Surprisingly, we also observed noticeable changes in the size and shape of those aggregates when incubated together, suggesting that the observed in vitro interaction might play a role in their in vivo interaction.

As previously mentioned, MSI proteins are predominantly expressed in NSCs[34]. Similarly we observed strong MSI1 expression in the SGZ of the hippocampi. MSI1-positive nuclei were found in the DG, CA1 and CA3 hippocampal regions, as well as many layers of the CTX. In our previous study using human brains, we observed MSI1 expression in mature human neurons[10] and confirmed this finding in mice, suggesting that MSI1 is expressed in fully differentiated and aging neuronal nuclei. In mice, we observed age-dependent change for both MSI1 and MSI2, but such change seemed more evident in MSI1. Furthermore, these different MSI1 and MSI2 cellular localizations, coupled with an incremental expression, are tau dependent.

MSI2 showed reduced signal and cytoplasmic distribution in the 7-months old P301L mice mainly in CA1, CA3, and CTX. MSI2 is expressed mainly in NSCs, but we observed MSI2-positive cells in CA1, CA3 and CTX, areas that are not considered canonical neurogenic regions. The general reduction observed in 7-months P301L mice can possibly be associated with a sub-population of MSI2 positive cells, which are not characterized in this study. A general reduction in aged mice can be due to the alteration of neurogenic cells that is already observed in this mouse models even with many controversies[35]. Similar to MSI1, nuclear MSI2 accumulation observed in human AD brain tissues suggests an altered cellular distribution of MSI2. However, further intensive study on MSI2 distribution in human neurons, as well as P301L mouse model, is required. In vitro observation showed that high levels of tau and its oligomeric form induce profound nuclear dysfunctions. Evidence on nuclear membrane deficiency and its association with neurodegeneration has previously been reported[32,36], which we supported with different observations in cells overexpressing a mutated form of tau. In particular, we observed a significant decrease of LaminB1 and LaminA/C

proteins levels in nuclear membrane instability. With it, we also observed a general down-regulation of nuclear membrane connectors, such as Emerin, LaminB Receptor, Man1 and Nesprin2. Chromatin connectors such as HP1 (heterochromatin marker), LAP-2 and BAF were also reduced.

These results might indicate that toxic tau affects most likely the nucleocytoplasmic import of NLS-cargos by preventing their nuclear entry through NPC and their release into the nucleus that requires RanGTP. As it was shown that toxic tau can co-aggregate with some FG-nucleoporins, e.g. with Nup98 or Nup62, and as such impact NPC permeability.

On the other hand, toxic tau is also compromising RanGTP/RanGDP gradient that would also result in the reduced amount of the NLS-cargo in the cell nuclei[27]. Consequently, an import of LaminA, LaminB1 and other lamina binding proteins into the nucleus might also be impaired and result in the impairment of lamina formation. Nup50 or Nup153 both have NLS-like sequences and can interact with nuclear transport receptors. We observed that both proteins were reduced in the nuclear fraction but significantly elevated in the cytoplasm, indicating that their recruitment during NPC assembly is hindered in the presence of pathogenic tau.

Furthermore, our data showed a decrease in the H3K4me3 levels, indicating an increase of the condensation state of chromatin. However, we did not observe significant changes in H3K9me3 (heterochromatin marker). Interestingly, we observed a marked reduction in the interactions between tau with LaminA, LaminB1 and Histone3, as well as with MSI1. Together, the data indicate that in the presence of high levels of oligomeric tau, MSI1 and non-pathological tau lost contact with the nuclear membrane and nucleosomes. At the same time, we observed an increase in the association of MSI2 and toxic tau conformers with Histone3 and Lamin proteins. These results were confirmed by MS analyses and suggest that with the increase of toxic oligomeric tau, MSI1 and non-pathological tau have reduced binding affinity for the nuclear membrane and nucleosomes, whereas those of toxic tau and MSI2 increase. These results suggest more intensive study with mouse models on possible transcriptional effects on MSI1 loss of function (LOF) and MSI2 GOF, and how this pathway affects both the transcription and pathogenesis of neurodegenerative diseases.

In conclusion, these observations suggest that high levels of tau in the presence of MSI plays a critical role in compromising nuclear activity and integrity. For this reason, identifying and elaborating upon the interplay of MSI/tau proteins could be the beginning of a frontier in the study of neurodegenerative diseases, which potentially may provide a basis for discovering therapeutic strategies.

## Methods

**Cell cultures and treatments.** Three cell lines were employed in this study: HEK-293, iHEK overexpressing WT tau and iHEK overexpressing mutated P301L tau. All three were maintained in Dulbecco's minimum essential medium (DMEM) supplemented with 10% fetal bovine serum (FBS) at 37 °C in 5% $CO_2$. To induce WT and mutant tau overexpression, iHEK cells were treated with 1 μg/ml tetracycline (Tet) for 24 h in FBS-depleted DMEM (Gibco$^{TM}$ LS11965118, Thermo Fisher Scientific). After 24 h incubation, cells were washed twice with medium to remove excess Tet, then collected and stained. Trypsin (Gibco$^{TM}$ Trypsin-EDTA (0.25%) phenol red (LS25200114 Thermo Fisher Scientific) was used added and incubated for 3 min to detach cells, and the solution was then centrifuged at $1000 \times g$ for 5 min. Cell pellets were collected and used for protein fractioning.

**Human tissue processing.** Postmortem frontal cortex tissue from subjects with AD ($N = 6$), FTD ($N = 6$), ALS ($N = 6$) and age-matched control ($N = 6$) were provided as frozen blocks by the Institute for Brain Aging and Dementia at UC Irvine after approval by the Institutional Ethics Committee. The samples were homogenized in 1× phosphate-buffered saline (PBS) mixed with a protease inhibitor cocktail (Roche) and phosphatase inhibitor (Sigma) at a 1:3 (w/v) dilution of brain tissue:PBS. Samples were then centrifuged at $11,000 \times g$ for 20 min at 4 °C. The supernatants (PBS-soluble fractions) were aliquoted, snap-frozen in liquid

nitrogen, and stored at −80 °C. The pellets (insoluble fractions) were resuspended in homogenization buffer (1× PBS), aliquoted and frozen at −80 °C until use.

**RNA extraction and RT-qPCR.** Total RNA from iHEK cells was extracted using TRIzol reagent using an established protocol[5]. RNA samples for real-time analysis were quantified using a Nanodrop Spectrophotometer (Nanodrop Technologies) and qualified on an RNA Nano chip with the Agilent 2100 Bioanalyzer (Agilent Technologies). Only samples with high-quality total RNA were used (RIN: 7.5–10.0). cDNA synthesis was performed with 0.5 or 1 μg of total RNA in a 20-μl reaction volume using Taqman Reverse Transcription Reagents Kits from Life Technologies (#N8080234). qPCR amplifications were performed in duplicate or triplicate using 1 μl of cDNA in a total volume of 20 μl using iTaq Universal SYBR Green Supermix (Bio-Rad #1725125). The final MAPT primer concentration was 300 nM. Relative RT-qPCR assays were performed using 18S RNA or another housekeeping gene for normalization. Absolute analyses were performed with known amounts of synthetic gene transcripts. PCR assays were carried out using the ABI Prism 7500 Sequence Detection System. The list of primers is shown in Supplementary Table 1.

**Animals.** Transgenic mice and control littermates were obtained from breeding colonies maintained at the University of Texas Medical Branch (UTMB) Animal Facility. All animals were genotyped from tail DNA (Gene Script). KO tau mice do not express tau. P301L tau mice (Jackson Laboratory, stock # 000664) overexpress human tau with a mutation in MAPT P301L, specifically the 4R/2N isoform, under control of the neuron-specific murine Thy1 promoter. Tau hyperphosphorylation and conformational changes in the brain begin at ~7 months. Tangle-like pathology affects the brainstem and spinal cord and to a lesser extent the midbrain and cerebral cortex[37]. Htau mice express all six isoforms of human wild-type tau but do not express mouse tau[38]. C57BL/6 mice were used as controls for the mouse tau model. The UTMB Institutional and Animal Care and Use Committee approved all animal protocols. Mice were housed in IACUC-approved vivarium at UTMB. We employed 2- and 7-month-old mice for immunofluorescent brain section studies. Prior to brain harvesting, all mice were perfused with 4%-paraformaldehyde in PBS, followed by cryoprotection in 30% (w/v) sucrose/PBS for 2–3 days at 4 °C. Sagittal brain sections for immunofluorescence were cut in 8-μm steps on a freezing microtome.

**Immunocytochemistry (ICC) and confocal microscopy.** Cells were placed on a 24-well coverslip, fixed with 0.5 ml of 4% paraformaldehyde/PBS for 15 min, then washed 3 times in PBS (5 min each). The cells were then permeabilized with 0.5 ml PBS/0.2% Triton X-100 (PBST) for 5 min and blocked in 0.5 ml of 5% NGS Serum in PBST for 1 h. The primary antibody was in diluted 5% normal goat serum (NGS)/PBST, and cells were incubated in this mixture overnight at 4 °C, then washed three times in PBST (10 min each). Secondary antibody (Thermo Fisher Scientific) was diluted 1;800 in 5% NGS/PBST, and cells were incubated for 2 h at room temperature (RT). After a wash, the nuclei were stained with DAPI diluted at 1:10,000 in PBST (5 mg/ml stock solution) for 5 min. Two washes were then performed (10 min each, PBST then PBS). Coverslips were mounted on glass slides with 8–10 μl Prolong Gold Antifade mounting media. The slides were air dried inside a fume hood at RT (or stored at 4 °C until they were placed in a fume hood). The primary antibodies used in this study for ICC were MSI1 (Abcam, ab52865 1:250), MSI2 (Abcam, ab76148 1 μg/ml) and Tau13 (1:200). After three PBS washes, cells were probed with mouse- and rabbit-specific fluorescent-labeled secondary antibodies (1:200, Alexa Fluor 488 and 546, Life Technologies). Single frame images and Z-stacks for 3D rendering and orthogonal view acquisition were collected using a Zeiss LSM880 confocal microscope equipped with a Nikon 63x oil immersion objective and processed with Zeiss Lite Black Software. Images for quantification of area, area ratio, integrated density, number of foci, and double extractions, were acquired on a Keyence BZ-800 microscope with a Nikon 100x oil immersion objective and assessed with a BZ-X Analyzer.

**Immunolabeling of fixed human and mouse brain sections.** Frozen human and mouse brain sections were first fixed in chilled 20% methanol then permeabilized in PBS/0.4% Triton-X100 for 5 min. After 1-h blocking in PBS/10%NGS/0.2% Triton at RT, the sections were incubated in primary antibodies in PBS/10% NGS overnight at 4 °C. Immunolabeling for MSI1 (1:250, ab52865, Abcam) and MSI2 (1:250, ab76148, Abcam) was performed in all three mouse models. The next day, sections were washed three times in PBS (5 min each), and secondary antibodies were applied as described for ICC: rabbit - Alexa Fluor 488 for MSI1 and MSI2 (1:200, Life Technologies). After further washing in PBS, the sections were incubated for 20 min with 0.1% (w/v) Sudan Black B solution to quench autofluorescence. The slides were then mounted with Prolong Gold Antifade with DAPI (Thermo Fisher Scientific P36931) to stain the nuclei.

**Proximity ligation assay (PLA).** Protein-protein interactions were identified using in situ proximity ligation assays. This approach relies on recognition of target molecules in close proximity (<40 nm) by affinity probe pairs that produce an amplifiable signal. PLAs in mouse brain tissue were carried out using Duolink® PLA in Situ Red starter kit mouse/rabbit (Sigma Aldrich, DUO92101) according to

the manufacturer's instructions. The incubation time and antibody concentrations were based on an immunofluorescence protocol. Primary antibodies used for in situ PLAs were: MSI1 (Abcam, ab52865 1:250), MSI2 (Abcam, ab76148 1 µg/ml), Tau13 (1:200), TOMA2 (1:200). The amplified red signal as detected with a Keyence microscope.

**Nuclear MSI/tau aggregate quantification in mouse neurons**. To quantify nuclear MSI foci, 100 nuclei from the control and treated groups were imaged and analyzed. We extracted the nuclear MSI signal from Z-stacks (0.5-µm step size) using the nuclear area as the target. To quantify the area, area ratio, density, number, diameter and fluorescence intensities of MSI1, MSI2, and T22 aggregates, we performed double extraction from the target areas using BZ-X Analyzer Software. The results were collected and graphed and analyzed using GraphPad Prism 6 Software.

**Immunofluorescence imaging analysis**. Images were captured with a 2.84 mp CCD Peltier Cooled Camera and stored for subsequent analyses. For 3D rendering (volumetric rendering), visualization and segmentation, Ten images per group were processed and segmented using Arivis Vision 4D Software. Analyses of nuclei and foci were performed on a BZ-X Analyzer from Keyence Company using 10 nuclei or regions of interest (ROIs) for each set of experiments. Area, area ratio, integrated density, and number and density of foci were measured after image thresholding and segmentation. Intensity profiles were produced from 10 cytoplasm ROIs and 10 nuclei for each condition. The measurements of all analyzed parameters were corrected for the elongation factor, which was estimated with 4-µm fluorescent nanospheres (TetraSpeck™ Fluorescent Microspheres Sampler Kit #T-7284, Thermo Fisher Scientific). GraphPad 7.0 Software was used to generate graphs and perform statistical analyses.

**Western blotting and compartment fractioning**. Western blot analyses were performed with iHEK cell lysates. Approximately 10 µg of proteins preparations were loaded on precast NuPAGE 4–12% Bis–Tris gels (Invitrogen) for sodium dodecyl sulfate polyacrylamide gel electrophoresis. Proteins were subsequently transferred onto nitrocellulose membranes and blocked overnight in 10% nonfat dry milk at 4 °C. Membranes were then probed for 1 h at RT using primary antibodies against α-MSI1 (1:1000, Abcam), α-MSI2 (1:1000, Abcam) and Pan-Tau (1:10,000, Tau13), GAPDH (1:1000), LaminB1 (1:1000) and Histone3 (1:1000) diluted in 5% nonfat dry milk. α- MSI1 and α-MSI2 immunoreactivities were detected with a horseradish peroxidase (HRP)-conjugated anti-rabbit IgG (1:6000, GE Healthcare). Immunoreactivity for other antibodies was detected using an anti-mouse IgG (1:6000, GE Healthcare) diluted in 5% nonfat milk. Enhanced chemi-luminescence (ECL) plus (GE Healthcare) was used for band visualization. LaminB1 and GAPDH were probed to normalize and quantify nuclear and cyto-plasmic proteins, respectively. Compartment extraction was conducted with Qproteome Cell Compartment Kits (Qiagen, #37502), and nuclear, cell membrane, and cytoplasmic proteins were isolated and preserved for western blot analysis.

**Separation of sarkosyl soluble and insoluble tau**. Soluble and insoluble tau were separated in using a well-established protocol[39]. Briefly, each brain tissue sample was homogenized in PBS containing protease inhibitors (Roche, Cat. # 11836145001) at a brain:PBS ratio of 1:3 (w/v) and incubated for 20 min at 4 °C. Samples were then centrifuged for 20 min at 11,000 × g at 4 °C. The supernatant was collected and centrifuged at 100,000 × g for 60 min at 4 °C. Pellets from the first and second cold centrifugations were combined, resuspended 1:10 (w/v) in PHF extraction buffer at 1:10 (w/v), and centrifuged for 20 min at 15,000 × g at 4 °C. Sarkosyl was added to the supernatant to a final concentration of 1% and incubated for 1 h at RT while stirring. The samples were then centrifuged for 30 min at 100,000 × g at 4 °C. Finally, the pellet was resuspended in TBS buffer (1 ml buffer for 25 g of starting material). To remove any monomer or oligomers trapped in the isolated PHF, samples were dialyzed against PBS using floating Spectra/Pro disposable dialysis devices with 100-kDA molecular weight cutoff membranes (Spectrum labs). Finally, three wash and spin-down (10,000 × g) cycles were run to remove any non-fibrillar materials before discarding the supernatant and resuspending the final pellet in PBS. Atomic force microscopy characterization and thioflavin T binding assays were performed. The samples were used as a pure brain-derived fibrillar tau (PHF).

**Filter trap assay**. Homogenate from AD, ALS, FTD, and control brains (~15 µg, soluble fraction) was applied onto pre-soaked (TBS-T) nitrocellulose membranes, using a vacuum-based bioslot apparatus using our published protocols[10,40]. The membranes were blocked with 10% nonfat dry milk in TBS-T solution overnight at 4 °C, then incubated at RT for 1 h with Tau13 (1:10,000), α-MSI1 (1:1000) and α-MSI2 (1:1000) antibodies diluted in 5% nonfat milk. Next, membranes were washed three times with TBS-T. Immunoreactivity was detected using an HRP-conjugated anti-mouse IgG (1:6000, GE Healthcare) or anti-rabbit IgG (1:6000, GE Healthcare), as appropriate. Membranes were washed three times in TBS-T prior to signal detection with ECL plus (GE Healthcare).

**Immunoprecipitation (IP)**. IP with the Toxic Tau Conformers antibody (TTC35) and MSI1 antibodies was conducted for the cytoplasmic and nuclear fractions of P301L iHEK cells. Cytoplasm/Nucleus Fractioning was described in the western blotting section. Pierce™ Co-Immunoprecipitation Kits (Thermo Scientific, #26149) were used according to manufacturer guidelines. Briefly, amine-reactive resin was coupled with MSI1 (Santa Cruz Biotechnology, [Y2709]) suitable for IP and affinity-purified TTC35 antibodies, then incubated with cytoplasmic and nuclear extracts. Bound proteins were eluted in 0.1 M glycine (pH 2.8), and the final pH was increased to 7.0 by adding 1 M Tris-HCl (pH 8). Isolated fractions were subjected to buffer exchange, collected in sterile PBS, and used for western blot analysis. Total protein concentrations were measured with a bicinchoninic acid protein assay (Micro BCA Kit, Pierce).

**Sample digestion for mass spectrometry (MS) analysis**. Agarose bead-bound purified proteins were washed several times with 50 mM TEAB pH 7.1 before solubilization with 40 µl of 5% SDS, 50 mM TEAB (pH 7.55) at RT for 30 min. The supernatant containing proteins of interest was then transferred to a new tube and reduced by titrating to 10 mM TCEP (Thermo Fisher Scientific, #77720) and incubating the mixture at 65 °C for 10 min. After samples cooled to RT, 3.75 µl of 1 M iodoacetamide acid was added and incubated for 20 min in the dark before 0.5 µl of 2 M DTT was added to quench the reaction. Next, 5 µl of 12% phosphoric acid was added to the 50 µl protein solution, followed by 350 µl of binding buffer (90% methanol, final TEAB concentration 100 mM; pH 7.1). The solution was then passed through an S-Trap spin column (protifi.com) using a bench-top centrifuge for 30 s at 4000 × g. The spin column was then washed with 400 µl of binding buffer and centrifuged three more times. Trypsin was added to the protein mixture at a 1:25 ratio in 50 mM TEAB (pH 8) and incubated for 4 h at 37 °C. Peptides were eluted with 80 µl of 50 mM TEAB, then 80 µl of 0.2% formic acid, and finally 80 µl of 50% acetonitrile/0.2% formic acid. The peptide solutions were combined, dried in a speed vac, resuspended in 2% acetonitrile/0.1% formic acid/97.9% water, and placed in autosampler vials.

**NanoLC MS/MS analysis**. Peptide mixtures were analyzed by nanoflow liquid chromatography-tandem mass spectrometry (nanoLC-MS/MS) on a nano-LC chromatography system (UltiMate 3000 RSLCnano, Dionex), coupled to a Thermo Orbitrap Fusion mass spectrometer (Thermo Fisher Scientific) through a nano-spray ion source (Thermo Scientific). A trap and elute method was used with a C18 PepMap100 trap column (300 µm × 5 mm, 5 µm particle size) from Thermo Fisher Scientific. The analytical column was an Acclaim PepMap 100 (75 µm × 25 cm, Thermo Fisher Scientific). Following column equilibration in 98% solvent, A (0.1% formic acid in water) and 2% solvent B (0.1% formic acid in acetonitrile), samples (2 µl in solvent A) were injected into the trap column and eluted (400 nl/min) by gradient elution onto the C18 column as follows: isocratic at 2% B, 0–5 min; 2 to 32% B, 5–39 min; 32–70% B, 39–49 min; 70–90% B, 49–50 min; isocratic at 90% B, 50–54 min; 90–2%, 54–55 min; and isocratic at 2% B, until 65 min.

All of the LC-MS/MS data were acquired with XCalibur, version 2.1.0 (Thermo Fisher Scientific) in the positive ion mode, and a top speed data-dependent acquisition method with a 3-s cycle time. The survey scans (m/z 350-1500) were acquired in the Orbitrap at a resolution of 120,000 (at m/z = 400) in profile mode, with a maximum injection time of 100 ms, automatic gain control (AGC) target of 400,000 ions, and S-lens RF level of 60. Isolation was performed in the quadrupole analyzer with a 1.6-Da isolation window, and CID MS/MS acquisition was carried out in the profile mode using a rapid scan rate with detection in the ion-trap using the following settings: parent threshold = 5000, collision energy = 32%, maximum injection time 56 ms, AGC target 500,000 ions. Monoisotopic precursor selection and charge state filtering were on, and charge states 2–6 were included. Dynamic exclusion was used to remove selected precursor ions (±10 ppm mass tolerance) for 15 s after acquisition of a single MS/MS spectrum.

**Database searching**. Tandem mass spectra were extracted, and the charge state was deconvoluted with Proteome Discoverer (Thermo Fisher, version 2.2.0388). Deisotoping was not performed. All MS/MS spectra were searched against the UniProt Human database (version 06-27-2018) using Sequest. The search parameters were parent and fragment ion tolerances of 5 ppm and 0.60 Da, respectively. Trypsin was specified as the enzyme, allowing for two missed cleavages. A fixed modification of carbamidomethyl and variable modifications of oxidation and glycosylation were specified in Sequest.

**Protein identification criteria**. Scaffold (version Scaffold_4.8.7, Proteome Software Inc.) was used to validate MS/MS-based peptide and protein identifications. Peptide identifications were accepted if they could be established at >95.0% probability using the Scaffold Local FDR algorithm. Protein identifications were accepted if they could be established at >99.0% probability and contained at least two identified peptides. Protein probabilities were assigned by the Protein Prophet algorithm[41]. Proteins that contained similar peptides and could not be differentiated with MS/MS analysis alone were grouped to satisfy parsimony principles. Proteins sharing significant peptide evidence were grouped in clusters.

**Tau oligomer production, labeling and cell treatments**. Recombinant tau oligomers were produced and characterized following a published protocol[42]. Tau oligomer labeling was performed as follows: 1 mg of Alexa Fluor™ 568 NHS Ester (Invitrogen) was dissolved in 0.1 M sodium bicarbonate to a final concentration of 1 mg/ml. The dye was incubated with Tau oligomers in a 1:2 (w/w) ratio. The mixture was rotated overnight at 4 °C on an orbital shaker. The next day, the solution was centrifuged (30 min, 15,000 × $g$) using a 10-kDa Amicon Ultra-0.5 Centrifugal Filter to remove unbound dye. The oligomers were then washed with 1× PBS until the flow-through solution was clear. The filter compartment was flipped and centrifuged to collect the concentrate, and the oligomers were reconstituted to their original volume. TauO-568 was resuspended in complete DMEM to obtain final solutions with 0.5 and 2 μM concentrations. The cells were treated with TauO for 1 h at 37 °C and 5% $CO_2$. Finally, the medium was removed, and the cells were collected for cytoplasmic and nuclear protein fractioning and immunofluorescence assays.

**Preparation of tau and MSI oligomers**. Glutathione-S-transferase-tagged MSI proteins were recombinantly expressed and purified, and proteins were aggregated following our standard protocol[10]. Briefly, both purified MSI1 and MSI2 proteins (0.5 μg/μl) were stirred on an orbital shaker with a Teflon-coated micro stir bars for 48 h inside the fume hood at RT in closed tubes. MSI1 and MSI2 oligomers were injected into a Shimadzu HPLC system fitted with a TSK-GEL G3000 SWXL (30 cm × 7.8 mm) column, Supelco-808541. The mobile phase was PBS (pH 7.4) with a flow rate of 0.5 ml/min. Calibrations were made using a gel filtration standard (Bio-Rad 51-1901).

**Atomic force microscopy (AFM)**. Oligomeric preparations of MSI1 and MSI2 proteins were subjected to AFM in ScanAsyst mode on a Multimode 8 AFM machine (Bruker). Briefly, 5-μl samples were applied onto a freshly cleaved mica surface and allowed to adsorb. The mica was then washed with 200 μl of deionized water and air dried.

**MSI1 silencing**. Cells ($2.5 \times 10^5$) were seeded in 6-well plates and incubated for 24 h with complete DMEM. After 24 h, MSI1 Gapmers (QIAGEN, LG00214872-EDB) transfections were carried out following with Lipofectamine 2000 (Thermo Fisher Scientific #11668–027) following the manufacturer's instructions using a final Gapmers concentration of 50 nM per well. After 5 min of RT incubation, the cells were incubated for 24 h with Lipofectamine/MSI1 Gapmers solution in DMEM without FBS. After 24 h, untreated and Gapmers-transfected cells were collected, and total lysates were prepared for western blot analysis of MSI1 and Tau13 (labeled Gapmer internalization was observed after 1 h of treatment).

**Statistical analysis and reproducibility**. All in vitro experiments were performed in at least triplicate. Micrographs from human and mouse tissue and cells were obtained from three independent experiments with similar results, and all figures show representative micrographs. All data are presented as means ± SD (standard deviation) and were analyzed using GraphPad Prism Software 6.0. Statistical analyses were carried out using Student's $t$ tests or one-way analyses of variance (ANOVAs) followed by Tukey's multiple comparisons testing. Column means were compared with one-way ANOVA using treatment as the independent variable. Group means were compared using two-way ANOVAs with factors on treatment. If ANOVA revealed a significant difference, pairwise comparisons between group means were performed with Tukey and Dunnett's multiple comparison tests.

**Reporting summary**. Further information on experimental design is available in the Nature Research Reporting Summary linked to this paper.

## Data availability

All relevant datasets used and/or analyzed in this current study are available from the corresponding author. Source data are provided with this paper.

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

## Acknowledgements

The authors thank the members of the Kayed lab for their support and help. We thank Dr. Bess Frost (University of Texas Medical Health Center, San Antonio) for her useful suggestions. This work was supported by Mitchell Center for Neurodegenerative Diseases, the Gillson Longenbaugh Foundation and National Institute of Health grants: R01AG054025, R01NS094557, RFA1AG055771, R01AG060718 and the American Heart Association collaborative grant 17CSA33620007 (R.K.). We thank Dr. Russell William (Mass Spec facility core in UTMB) for his help and assistance with mass spectrometry analysis. Dr. Ivannikov Maxim for microscopy assistance (Microscopy Facility Core in UTMB). We thank Claudia Di Gesu' and other trainees for helping with immunofluorescence.

## Author contributions

Conceptualization, M.M. and R.K.; Methodology, M.M., and R.K.; Investigation, M.M., S.M., N.P., N.B.,U.S., O.J., M.G.K. and R.K.; Writing—Original Draft, M.M.; Writing—Review & Editing, all authors; Funding Acquisition, R.K.; Resources, R.K.; Supervision, M.M. and R.K.

## Competing interests

The authors declare no competing interests.
