## [Peer Review File · Nature Communications]

Reviewers' comments:

Reviewer #1 (Remarks to the Author):

- The premise of the manuscript is the modulation of tau neurotoxicity by Musashi (MS) proteins. The finding of the role of RNA-binding proteins Musashi on tau soluble aggregates is novel. Indeed, aggregate Tau-mediated toxicity is relevant to understand the pathogenesis of AD, and therefore, the groups finding of MS-initiated nuclear dysfunction in tauopathies assumes significance.
- Comparing MSI protein accumulation between different tauopathies allows for the possible identification of distinctive and/or common pathogenic mechanisms between AD and other neurodegenerative diseases, and hence they will be of interest to others in the community and the wider field.
- In synopsis, the present work is a well-designed and well-executed study. The manuscript is well-written, and presentations of different points and discussion would move the field forward.

Criticism:

- Fig. 1-I- MS1- How many immunoreactive bands did appear in WB when probed with anti-MS1 antibody? A full blot picture of anti-MS1 with HEK cell lysate and brain extracts should be shown. The Suppl fig shows only the activity of the secondary antibody alone.
- Validation of the specific anti-MS1 band(s) by siRNA-type experiment would be more specific and is advisable, if possible.
- Fig. 1-J- MS2- the same criticisms as above exist.

-Fig. 2: Authors should be more specific about the human brain region tested. Which Brodmann area (number) was used? This may be relevant/ important for studying human brain disorders (different than animal work).

-Fig. 2-F &G- Number of human brain tissues are very limited. This reviewer understands the difficulty in procuring autopsied samples from well-matched AD and control subjects; however, the study seems underpower, particularly from the context of several confounding factors.

- I think there are inconsistencies regarding human brain samples. Table 1 shows 11 samples, ALS n=3, FTD n=2, AD n=3, control n=3. The blots indicate 12 samples, n=3 for each condition.

-Fig. 6D- In addition to "Control", it would be better to show "Control+Tet" treatment in parallel.

-Supplemental Figure 1. Which region of the "brain" in brain homogenates were used?

-Further, have the authors anything to say about the stark difference between AD and other samples regarding A11 blot responses?

-The cytoplasmic/nuclear blots require a bit more clarity. Do the authors have any probeds that show a response in the cytoplasmic samples for any protein at all?

General comments: Is PCC abbreviation, used in the text, standard?. Readers may confuse it with Posterior Cingulate Cortex over Pearson's Correlation Coefficient.

Reviewer #2 (Remarks to the Author):

Summary

The submitted article by Montalbano et al. observes the interaction of oligomeric tau with the RNA binding proteins musashi 1 and 2 in several neurodegenerative diseases, mouse models and stable transgenic cell lines. There are the makings of a very interesting finding here. However, at present there is disjointed flow to the data and some over-reach in the interpretation of the result meaning the conclusions are a little premature.

It is interesting to see that MSI1 and 2 are elevated in neurodegenerative disease – but it is not

clear how this is related to "tauopathies". The strength of the manuscript suffers because the colocalization with tau oligomers appears to be low and the authors appear to have similar elevation of MSI1 and 2 in ALS and FTD without tau tangles, perhaps suggesting this elevation is a non-specific effect of neurodegenerative disease. It is interesting to see that there are nuclear membrane and chromatin modelling deficits in the inducible tau cell line – but it is not clear that this is directly related to formation of "mushashi and tau soluble aggregates".

Major comments

The rationale for studying MSI 1 and 2 in tauopathies appears to be supported by the previous work of this group and others. However, the rationale for studying tau oligomer/MSI interaction in ALS and FTD is less apparent. This issue is particularly relevant to the question of whether mushashi dysfunction occur specifically in response to tau pathology or non-specifically in response to neurodegenerative disease. The FTD cases used in this study are poorly defined. Table 1 indicates that the FTD cases are clinically diagnosed but have pathological observation of minimal Braak NFTs (less in fact than the control individuals) and little tau accumulation by immunoblot (Supp Figure 1B), which would suggest these FTD cases present with TDP-43 pathology. There should be a neuropathological diagnosis of either FTLT-tau or FTLT-TDP. Furthermore, tau pathology is only associated with a small minority of ALS cases. This naturally raises the question of whether MSI1 and 2 colocalize and interact with the major pathology of ALS and FTLT-TDP, which is TDP-43. This seems particularly pertinent as there appears to be minimal colocalization of TOMA2 oligomeric tau and MSI1 and 2 (Figure 1G, H).

It appears that the microscopic images of TOMA2 and MSI1 and 2 distribution do not clearly support the author's assertion that TOMA2 oligomeric tau and MSI1 and 2 colocalise in AD, ALS and FTD. In figure 1G, H, in figure 2C, D, in supplemental figure 2, in supplemental figure 7 the vast majority of pixels are either red (TOMA2) or green (MSI1 or 2) but very little is yellow = colocalization. Indeed, the TOMA 2 signal in control tissue (supplemental figure 2) appears to be minimal, which may explain why an increase in colocalization between TOMA2 and MSIs is observed in the affected patient tissue. But it doesn't follow that the TOMA2 and MSIs actively associate. In the patient samples (particularly the ALS and FTD samples) there appears to be accumulation of both TOMA2 and MSIs, but these appear to be in distinct puncta.

On the other hand, in figure 2B there does appear to be a higher degree of colocalization of TOMA2 and MSI1, and this is supported by the biochemical association of tau MSI1 and 2 with tau in the AD insoluble fractions (something which does not occur in the ALS and FTD cases). Line 454-455 – "...colocalization of MSI proteins and tau oligomers play a crucial role in AD, ALS and FTD..."

The authors indicate (line 573 to 575) that higher mag images of tau transgenic mouse tau13/MSI PLA "allows us to appreciate... extracellular speckles, present in CTX, DG, CA1 and CA3" - but the panels in figure 5C to J are too low magnification to distinguish intra and extracellular distribution. In addition, the double labeling for TOMA2/MSI presented in Suppl. Fig. 7 (note that the text wrongly refers to Suppl. Fig. 6), does not present convincing evidence of double labeling, contrary to what is stated in the results section.

There appears to be a somewhat hard transition to the data of figure 6. Whereas the prior figures focused on mouse brain tissues, figures 6 and 7 focus only on iHEK cells, with no reference to or validation in brain tissues. Thus, the flow of the article seems to revert to additional figures from the author's previous publication "Tau oligomers mediate aggregation of RNA-binding proteins Musashi1 and Musashi2 inducing Lamin alteration" (Aging Cell, 2019). While this is excellent data it would need to be validated in mouse brain tissue to represent a significant conceptual advance from the previous work. At the very least, it would be prudent to test whether tau and MSI oligomers cause nuclear membrane defects: show interaction or tau oligomers and/or MSI oligomers with nuclear membrane components, show that addition of MSI oligomers alone induce nuclear membrane defects or show that tau-oligomer induced nuclear membrane defects are blocked by modifying MSIs. Indeed, the from the IPs of MSI1 and 2 show mostly interaction with histones, which is a really interesting finding with perhaps significant effects on gene expression. However, again the work is about mushashi and tau soluble aggregates [initiating] nuclear dysfunction", but there is little evidence herein that it is specifically soluble oligomers that are exerting these effects in the iHEK cells.

Minor comments

References 27 and 28 are duplicated. Line 111.

Line 132 – Would researchers in the field often refer to ALS and FTD without NFTs as tauopathies?

Miss-spelling of Histone3. Line 246

Line 442 – “Figure 1F-1G” appears to be incorrect

Please include high mag images from TOMA2 and MSI1 staining of control tissue in main figure 2.

Line 488 – the authors presumably mean “aged P301L mice.”

Line 492 – there appears to be no figure 3S (perhaps this is a typo).

Figure 3H and K – could the authors make the white/red arrows a little easier to see.

Figure 3 legend – several miss-spellings of “month”. Lines 515, 523 and 525.

Line 540 – “we detected a cell specific staining in 7-month old mice”. Which specific cell types are the authors referring to? Was staining with cell-type markers performed?

The authors might discuss why MSI2 shows reduced signal and cytoplasmic distribution in the 7 month old P301L mice (CA1, CA3, CTX) (line 543-544), but increased signal and perhaps nuclear distribution in AD (line 397).

There is no discussion of supplemental figures 4 and 5 in the main text. Please discuss how aging in these WT and tauKO mice affects MSI1 and 2; and whether the changes in MSI1 and 2 in figure 4 are specific to the tau expression. As such there it is unclear that changes in MSI are tau-dependent (line 748).

Line 566 to 568 – Contradicts figure 4 showing increased nuclear signal from MSI1 in 2 month old P301L mice? Line 532 “In general, we detected in P301L 2-months old mice a nuclear and axonal MSI1 distribution in hippocampi and cortex”.

Please include secondary only controls to confirm specificity of antibodies for IHC (and either a citation or use of knockout tissue to confirm TOMA2 specificity).

Figure 5 – in 2month old P301L mice, the low magnification image panel A shows no MSI1/TOMA2 PLA signal in DG. Higher mag image panel D shows robust MSI1/TOMA2 PLA signal in DG nuclei.

Could the authors explain this apparent mismatch.

Supplemental Figure 6: The text states that the TOMA2/MSI immunofluorescence is in

Supplemental figure 6, but it is actually in Supplemental Figure 7.

Line 648 – It appears that removal of tet does lead to cells without nuclear membrane defects. The 24hr on, 48hr off experimental condition leads to much fewer cells. Is it possible that those cells that had nuclear membrane dysregulation are dying, perhaps indicating poor recovery even after removal of tet?

Line 670 – “For euchromatin, we used H3K4me3 and H3K9me3 for heterochromatin”. This phrasing is confusing, which marker was used for which chromatin state?

Lines 698 to 700 – it is hard to follow the meaning of: “Furthermore, TTC implement interaction (after Tet induction) with LaminB1, while MSI1, as well as for the Histones, showed a marked reduction of interaction (Figure 7I).”

There is often use of terms that are somewhat unconventional. The authors should more clearly define their use of the following terms:

- What is a “slot plot”. Line 458. Used in figures 2F, G and supp figure 3C.

- “LUT”. Used in figures 3H, 7D

- “increment” and “decrement” (throughout the manuscript, examples: line 671, 673, 684, 696 etc)

o Usually defined at the “quantity of an increase”

- Line 696 - “TTC35” – no indication in text as to what this is.

- “membrane bumping” line 647.

Reviewer #3 (Remarks to the Author):

The study by Montalbano et al. demonstrates convincingly that tau oligomers can interact and form soluble aggregates with MSI1 and MSI2 proteins in vitro (HEK cells) and in the mature neurons.

Authors suggest that the pathogenic tau forms such aggregates in the neuron nuclei of the AD, ALS and FTD human and mouse brains that leads to the remodeling of chromatin packing and nuclear membrane structure. Co-localization of MSI1 and MSI2 with tau bundles is also observed in iHEK cells or mouse brain, where the expression of the P301L tau mutant was induced. Latter is known to form similar to the pathogenic tau aggregates.

The results show also that MSI1 is accumulating in nuclei of the aged P301L mice (7 month), but not associated to nucleoli and is not homogeneously distributed. MSI2 is low expressed in 2-month-old mice, but increases with age and detected in neuronal progenitor stem cells and mainly cytoplasmic.

After the performing different co-IPs and the mass-spectroscopic analysis of the nuclear and cytoplasmic fraction (see Fig. 7), the authors conclude that folded nuclear tau and MSI1 regulate the nuclear activity and stabilize chromatin distribution. In contrast, toxic tau conformers interact with MSI2 and destabilize the nuclear envelope, nucleoporins and chromatin localization.

They suggest also that the pathogenic tau interaction with MSI1 and MSI2 proteins in the cell nuclei of the mature neurons can impact nucleocytoplasmic transport and cause lamina disfunction.

All these observations are novel and would be highly interesting for the community focused on the neurodegenerative diseases. This work might be of the great interest for the wider field of scientist as well.

Particularly, I appreciate their extensive mass spectroscopic analysis combined with co-IPs and IF that provides valuable information on how different nucleoporins of nuclear pore complexes (NPCs), importins and NLS-cargos (e.g. different histones, laminA, laminB1, emerin, LAPs etc.) are affected by the accumulation of the toxic/pathogenic tau in the cells.

As shown in Fig. 6 F-H or in Fig. S8 D most of the cargo proteins that bare nuclear localization signal (NLS) are significantly reduced in the nucleus after the Tet-induced expression of toxic P301L tau. It might indicate that the toxic tau affects most likely a nucleocytoplasmic import of these NLS-cargos by preventing their nuclear entry through NPC and release into the nucleus that requires RanGTP. As it was shown previously (Eftekharzadeh et. al, 2018, 2019) toxic tau can co-aggregate with some FG-nucleoporins, e.g. with Nup98 or Nup62, and as such impact NPC permeability. On the other hand, toxic tau is also compromising RanGTP/RanGDP gradient (Eftekharzadeh et. al, 2018, 2019) that would also result in the reduced amount of the NLS-cargo in the cell nuclei. Consequently, an import of LaminA, LaminB1 and other lamina binding proteins into the nucleus might also be impaired and result in the mis-formation of the proper lamina. Nup50 or Nup153 (both bare NLS-signal and are delivered into the nucleus by importins during NPC assembly with following incorporation into the nuclear basket) are also reduced in the nuclear fraction and significantly elevated in the cytoplasm that might mean that their delivery into the nucleus is also hindered (but not cellular expression level). It would be beneficial for the paper, if these considerations are also taken into account in the discussion part.

I would highly recommend this paper for the publication in Nature Communication after the minor revisions.

Minor:

1. Lane 111: I guess here instead '...through binding of the nuclear pore complex (NUP), Nup98.' It should be written: '...through binding of the nuclear pore complex (NPC), e.g. to Nup98'.
2. Lane 128: does it mean here 'P301L' tau mutant (instead of 'P310L')?
3. Lane 376: and Fig. 1 (G and H) aggregates of tau and MSI 1 and MSI 2 are shown respectively. Authors claim that tau and MSI protein aggregates are co-localizing that is not really obvious from

these IF figures.

4. Fig. S3 D and F: should it be 'MSI2' written in green in the IF headings?

5. Fig 5: when comparing panels C and G; D and H; E and F; and F and G, it seems logic to conclude that the difference in PLA staining between upper and lower panels (e.g. in C) is due to the tau-13 and TOMA-2 staining, because second component (MSI1 antibody is present in both panels). Therefore, it is expected that the same difference in staining should be observed in PLA panels (low versus top) in G (same age, same mice model; same brain area) with extra signal that could be observed (or not) for MSI2 in this case instead of MSI1 like in C. But it seems PLA signal is much weaker here. Why? And why it is interpreted as difference in the MSI1 and MSI2 localization? Same question, when comparing panels D and H in this figure.

6. Fig. S7: it is shown that tau is enveloping nuclei, but not really present inside of nuclei as much, but MSI1 is mainly nuclear localized in P301L mice brain. Why is it concluded that they are co-localized in the nucleus?

7. Lane 627: LaminB1 and histone 3 level is decreased in the 7 months old P301L mice. Could it be connected with the disruption of NCT (nucleocytoplasmic transport), blockage of NPCs upon oligomeric toxic tau accumulation in the aged mice? If yes, this assumption would be then consistent with the mass spect. data shown in Fig. 6. It seems that nuclear import of NLS-type cargos is impaired and LAPs, LaminA and B and some basket NPC proteins like Nup153, Nup50 are reduced in the nuclear fraction and their amount elevated in the cytoplasm This could be really an indication that their expression level is not much affected, but their nuclear import is partly inhibited by the aggregating tau.

8. Lane 635: might be better to write 'nuclear proteins' instead of 'receptors';

9. Lane 638: 'nucleoporins' instead of 'Nucleoporin' in singular.

10. Fig. 7: IP in Fig. 7G shows that toxic tau interacts with histones directly or via MSI2 (if considering Fig.7 E and G). For stronger conclusion (shown in Fig. 7 J,K) a direct interaction of toxic tau (e.g. P301L mutant tau) with the recombinant MSI1 and MSI2 or with histones (e.g. histone 3) and lamins would be a plus. It would help to clarify whether toxic tau reduces interaction of MSI1 with histones by binding to them in the nucleus (Fig. 7 H) or less histones are detected in the nuclear fraction of this IP only because histones less efficiently delivered into the nucleus after Tet-induced P301L tau expression (due to change in pore permeability, interaction with tau at the nuclear pores etc.).

Authors' response to the reviewers

We would like to thank the reviewers for their important comments and suggestions.

Point-by-point response to the reviewers' comments

Reviewers' comments:

Reviewer #1 (Remarks to the Author):

-The premise of the manuscript is the modulation of tau neurotoxicity by Musashi (MS) proteins. The finding of the role of RNA-binding proteins Musashi on tau soluble aggregates is novel. Indeed, aggregate Tau-mediated toxicity is relevant to understand the pathogenesis of AD, and therefore, the groups finding of MS-initiated nuclear dysfunction in tauopathies assumes significance.

-Comparing MSI protein accumulation between different tauopathies allows for the possible identification of distinctive and/or common pathogenic mechanisms between AD and other neurodegenerative diseases, and hence they will be of interest to others in the community and the wider field.

-In synopsis, the present work is a well-designed and well-executed study. The manuscript is well-written, and presentations of different points and discussion would move the field forward.

Thank you so much!

Criticism:

Fig. 1-I- MS1- How many immunoreactive bands did appear in WB when probed with anti-MS1 antibody? A full blot picture of anti-MS1 with HEK cell lysate and brain extracts should be shown. The Suppl fig shows only the activity of the secondary antibody alone.

Validation of the specific anti-MS1 band(s) by siRNA-type experiment would be more specific and is advisable, if possible.

Fig. 1-J- MS2- the same criticisms as above exist.

Response: Thank you! We have added a full blot with HEK cell total lysate (Ctr and +Tet) and brain extracts with the anti-MSI1, anti-MSI2 in **Supplementary Figure 1h-i**. We performed in our previous study, published in *Aging Cell* [1], silencing-RNA MSI1 experiment using LNA Gappers technology (antisense oligonucleotides that provide highly efficient knockdown of mRNA) and we verified MSI1 silencing using the same primary antibody for MSI1 used in this study. As requested, we repeated it and performed a new MSI1 silencing in P301L tau iHEK cells (using MSI1 Gapmer) and added the results in **Supplementary Figure 1d-f**). We have also added a paragraph on MSI1 silencing by Gappers in the **Materials and Methods** section.

#Fig. 2: Authors should be more specific about the human brain region tested. Which Brodmann area (number) was used? This may be relevant/ important for studying human brain disorders (different than animal work).

Response: Thank you for this great suggestion. We have studied frontal cortices of the brain tissues and have added this information in **Figure 1**, **Figure 2** and **Supplementary Figure 1** legends. A new section “Human brain subjects and tissue harvesting” has been included in **Materials and Methods** section.

“Human brain subjects and Tissue harvesting. Frontal cortices of frozen brain tissues from age-matched control subjects (N=6), AD cases (N=6), FTD cases (N=6) and ALS cases (N=6) were received as frozen blocks from the Institute for Brain Aging and Dementia at UC Irvine, approved by the Institutional Ethics Committee. Brain tissues were homogenized in 1 X PBS mixed with a protease inhibitor cocktail (Roche) and phosphatase inhibitor (Sigma) at 1:3 (w/v) dilution of brain: PBS. Samples were then centrifuged at 10,000 rpm for 20 min at 4°C. The supernatants, PBS-soluble fractions were aliquoted, snap-frozen, and stored at -80°C until use. The pellets were resuspended in the homogenization buffer (1 X PBS) and were considered as insoluble fractions. They were also aliquoted and frozen at -80°C until use”. (Page 5) The detail information about the Brodmann areas were not provided to us.

#Fig. 2-F &G- Number of human brain tissues are very limited. This reviewer understands the difficulty in procuring autopsied samples from well-matched AD and control subjects; however, the study seems underpower, particularly from the context of several confounding factors.

- I think there are inconsistencies regarding human brain samples. Table 1 shows 11 samples, ALS n=3, FTD n=2, AD n=3, control n=3. The blots indicate 12 samples, n=3 for each condition.

Response: Thank you! There was a typo error in the original table, which is now corrected. As suggested by the reviewer, we have doubled the number of each pathology (AD, N=6; ALS, N=6; FTD, N=6 and age-matched controls, N=6) for analyses that are included in revised Table 1.

#Fig. 6D- In addition to “Control”, it would be better to show “Control+Tet” treatment in parallel.

Response: Thank you! We apologize for the confusion. Now we clarified it in the text ‘Control’ means not treated with Tetracycline (-Tet) and without tau induction.

Supplemental Figure 1. Which region of the “brain” in brain homogenates were used?

Response: We have added ‘frontal cortex’ in the **Supplementary Figure 1** legend. Thank you!

#Further, have the authors anything to say about the stark difference between AD and other samples regarding A11 blot responses?

Response: A11 antibody is an anti-oligomeric antibody (it does not recognize amyloid monomers and fibrils) [2]. We have also studied the oligomeric nature of Musashi proteins in AD using A11 antibody in our previous study [3]. In addition to the MSI oligomers, A11 also recognizes A β oligomers that are expected to be abundant in the AD brains and α -Synuclein oligomers [4]. Therefore, the observation of the stark differences between AD and the other pathologies was as AD brain tissues exhibit abundant oligomers of several amyloid proteins than other pathologies used in this study.

#The cytoplasmic/nuclear blots require a bit more clarity. Do the authors have any probing that show a response in the cytoplasmic samples for any protein at all?

Response: Answer: Thank you, we added GAPDH control in **Figure 6d** and **Figure 7a, 7b, 7e, 7f** and **7g**.

General comments: Is PCC abbreviation, used in the text, standard? Readers may confuse it with Posterior Cingulate Cortex over Pearson's Correlation Coefficient.

Response: Thank you so much! Now we abbreviated correctly.

Reviewer #2 (Remarks to the Author):

Summary

The submitted article by Montalbano et al. observes the interaction of oligomeric tau with the RNA binding proteins mshashi 1 and 2 in several neurodegenerative diseases, mouse models and stable transgenic cell lines. There are the makings of a very interesting finding here. However, at present there is disjointed flow to the data and some over-reach in the interpretation of the result meaning the conclusions are a little premature.

It is interesting to see that MSI1 and 2 are elevated in neurodegenerative disease – but it is not clear how this is related to “tauopathies”. The strength of the manuscript suffers because the colocalization with tau oligomers appears to be low and the authors appear to have similar elevation of MSI1 and 2 in ALS and FTD without tau tangles, perhaps suggesting this elevation is a non-specific effect of neurodegenerative disease. It is interesting to see that there are nuclear membrane and chromatin modelling deficits in the inducible tau cell line – but it is not clear that this is directly related to formation of “mshashi and tau soluble aggregates”.

Thank you!

Major comments

The rationale for studying MSI 1 and 2 in tauopathies appears to be supported by the previous work of this group and others. However, the rationale for studying tau oligomer/MSI interaction in ALS and FTD is less apparent. This issue is particularly relevant to the question of whether mshashi dysfunction occur specifically in response to tau pathology or non-specifically in response to neurodegenerative disease. The FTD cases used in this study are poorly defined. Table 1 indicates that the FTD cases are clinically diagnosed but have pathological observation of minimal Braak NFTs (less in fact than the control individuals) and little tau accumulation by immunoblot (Supp Figure 1B), which would suggest these FTD cases present with TDP-43 pathology. There should be a neuropathological diagnosis of either FTLT-tau or FTLT-TDP. Furthermore, tau pathology is only associated with a small minority of ALS cases. This naturally raises the question of whether MSI1 and 2 colocalize and interact with the major pathology of ALS and FTLT-TDP, which is TDP-43. This seems particularly pertinent as there appears to be minimal colocalization of TOMA2 oligomeric tau and MSI1 and 2 (Figure 1G, H).

Response: Thank you for your comment. As the expert reviewer noted FTLT patients have depositions of different misfolded proteins and recently FTLTs brains are subtyped in three major broad categories defined by intracellular protein inclusions: TDP-43 (FTLT-TDP), tau (FTLT-tau) and fused in sarcoma protein (FTLT-FUS)[5]. The cases included in the study were not sub-grouped the FTD cases evaluating TDP-43, FUS and tau accumulation in these brains because we could not obtain tissues. As suggested by the reviewer that they possibly are TDP-43 related cases. We did not measure the amount and/or the accumulation of TDP-43 and FUS

since this must be done by expert pathologist who collect and classify brains using standard brain bank methods. Our aim was evaluating MSI oligomers and interactions with tau in the tissues from the three different diseases. Since tau oligomers were not evaluated in the FTD subgroups; for years we tried, and were unsuccessful to obtain these tissues. Recently we tried again and reached out to two NIH funded brain banks requesting for cases from each subgroup, so far, they did not provide them. We hope this paper will make them more interested and perhaps they will investigate this in large well-characterized cohorts. From our side we are planning to get in touch with UPenn, Mayo Clinic and UCSF and suggest looking at these interactions in their cohorts, by offering our reagent and expertise as needed. We hope this will be done in the near future.

#It appears that the microscopic images of TOMA2 and MSI1 and 2 distribution do not clearly support the author's assertion that TOMA2 oligomeric tau and MSI1 and 2 colocalise in AD, ALS and FTD. In figure 1G, H, in figure 2C, D, in supplemental figure 2, in supplemental figure 7 the vast majority of pixels are either red (TOMA2) or green (MSI1 or 2) but very little is yellow = colocalization. Indeed, the TOMA 2 signal in control tissue (supplemental figure 2) appears to be minimal, which may explain why an increase in colocalization between TOMA2 and MSIs is observed in the affected patient tissue. But it doesn't follow that the TOMA2 and MSIs actively associate. In the patient samples (particularly the ALS and FTD samples) there appears to be accumulation of both TOMA2 and MSIs, but these appear to be in distinct puncta.

Response: Thank you for this observation! In **Figure 1g** and **h** we did not show quantification of colocalization because as noticed by reviewer we did not observe strong overlap (between green and red fluorescence) as indicated by the weak yellow signal. Therefore, we have chosen to show differential regions of interest (ROIs) from the same patients in **Figure 2**. As shown in **Figure 1g, h**, we observed intra-subject variability. Since the colocalization quantification (PCC) could underestimate co-localization of Musashi proteins with TOMA-2, therefore, we used regions of interests for a more precise estimation of colocalization. We also agree with the reviewer's comment that even though we have a positive correlation, the two protein markers do not colocalize completely. However, we also cannot exclude the partial overlap as shown by the ROIs positive correlation (**Supplementary Figure 2**). As described in ALS and FTD the TOMA-2 immunoreactivity is very different with numerous puncta in ALS and more compact in FTD, in accordance with recent studies demonstrating that different diseases have a distinct tau deposits - strains [6], [7]. As requested, we have now doubled the number of cases from each pathology (6 for each group), studied in **Supplementary Figure 2**.

On the other hand, in figure 2B there does appear to be a higher degree of colocalization of TOMA2 and MSI1, and this is supported by the biochemical association of tau MSI1 and 2 with tau in the AD insoluble fractions (something which does not occur in the ALS and FTD cases). Line 454-455 – "...colocalization of MSI proteins and tau oligomers play a crucial role in AD, ALS and FTD..."

Response: Corrected, thank you! In **Figure 2f**, we compared the PCC between the 4 groups with significant difference between control and diseased brains. We have re-phrased the sentence as below:

“These observations indicate that the interactions and colocalization of MSI proteins with tau oligomers play a crucial role in AD, ALS and FTD with different grades, and that probably elicit different effects on the cytoplasm and nuclei of neurons” (Page 14)

The authors indicate (line 573 to 575) that higher mag images of tau transgenic mouse tau13/MSI PLA “allows us to appreciate... extracellular speckles, present in CTX, DG, CA1 and CA3” - but the panels in figure 5C to J are too low magnification to distinguish intra and extracellular distribution. In addition, the double labeling for TOMA2/MSI presented in Suppl. Fig. 7 (note that the text wrongly refers to Suppl. Fig. 6), does not present convincing evidence of double labeling, contrary to what is stated in the results section.

Response: As suggested, we removed “extracellular distribution” from **Figure 5c-j**. In **Figure Supplemental 7**, we co-stained MSI1 (green) and TOMA-2 (red) and we observed a partial overlap of their signals, which suggest their co-presence in some cells. This observation indicates that in the cortex of aged P301L mice, MSI1+/TOMA-2- MSI1-/TOMA-2+ and double positive co-exist in different cell populations. To point out these differences we placed arrows on the merged channels to identify the three phenotypes. The same observation holds for co-staining with MSI1 and Tau13.

Thank you! Numbers for **Supplementary Figure 6** and **Supplementary Figure 7** have been corrected in the text.

There appears to be a somewhat hard transition to the data of figure 6. Whereas the prior figures focused on mouse brain tissues, figures 6 and 7 focus only on iHEK cells, with no reference to or validation in brain tissues. Thus, the flow of the article seems to revert to additional figures from the author’s previous publication “Tau oligomers mediate aggregation of RNA-binding proteins Musashi1 and Musashi2 inducing Lamin alteration” (Aging Cell, 2019). While this is excellent data it would need to be validated in mouse brain tissue to represent a significant conceptual advance from the previous work.

Response: Thank you for this excellent suggestion/comment! We have added the Western blot images of 2- and 7-months old mouse brain homogenates from C57, Tau KO and Tau P301L in **Figure 4**. Immunoblots for MSI1, MSI2 and GAPDH have been performed and added. In **Figure 4**, also we presented the relative density of monomeric MSI1 (mMSI1), total MSI1 along with monomeric MSI2 (mMSI2) and total MSI2.

#At the very least, it would be prudent to test whether tau and MSI oligomers cause nuclear membrane defects: show interaction or tau oligomers and/or MSI oligomers with nuclear membrane components, show that addition of MSI oligomers alone induce nuclear membrane defects or show that tau-oligomer induced nuclear membrane defects are blocked by modifying MSIs. Indeed, the from the IPs of MSI1 and 2 show mostly interaction with histones, which is a

really interesting finding with perhaps significant effects on gene expression. However, again the work is about mushashi and tau soluble aggregates [initiating] nuclear dysfunction”, but there is little evidence herein that it is specifically soluble oligomers that are exerting these effects in the iHEK cells.

Response: Thank you! We have done some experiments with MSI oligomers in our previous publications [3] and plan to continue, hopefully in the near future, by following this excellent suggestion. At this time, we could not because we receive the high-quality MSI proteins from our longtime collaborator and coauthor Dr. Kharas. Due to the current pandemic Covid-19 situation, we are unable to receive any MSI proteins from Dr. Kharas’s lab right now. We apologize for this inconvenience and thank you for your consideration.

Minor comments

#References 27 and 28 are duplicated. Line 111. - Corrected and Updated, Thank you!

Line 132 – Would researchers in the field often refer to ALS and FTD without NFTs as tauopathies?

Response: Thanks, we totally agree with you. It is a hard one and the field is still trying to figure it out. We cannot claim it is a tauopathy until pathologists classify it so. Therefore, we removed the word tauopathies from the title.

Miss-spelling of Histone3. Line 246 –

Response: Corrected, Thanks.

Line 442 – “Figure 1F-1G” appears to be incorrect,

Response: Corrected to **Figure 1g-1h**”. Thank you!

Please include high mag images from TOMA2 and MS11 staining of control tissue in main figure 2.

Response: Thank you! Now we included in, **Figure 2b**, representative images from TOMA2/MS11 staining of control tissue with scatter plot and colocalization pixel map.

Line 488 – the authors presumably mean “aged P301L mice.

Response: Corrected. Thank you

Line 492 – there appears to be no figure 3S (perhaps this is a typo).

Figure 3H and K – could the authors make the white/red arrows a little easier to see.

Response: Thank you so much! Modified as suggested.

Figure 3 legend – several miss-spellings of “month”. Lines 515, 523 and 525.

Response: Corrected- Thanks

Line 540 – “we detected a cell specific staining in 7-month old mice”. Which specific cell types are the authors referring to? Was staining with cell-type markers performed?

Response: Thank you. We have re-phrased the sentence (Page 18-19). We adjusted it as follows: Great point. The morphology and the position of the MSI2-positive cells strongly suggest their staminal nature. This finding is also supported by the fact that MSI2 is strongly expressed in the neuronal stem cells [8].

The authors might discuss why MSI2 shows reduced signal and cytoplasmic distribution in the 7-month-old P301L mice (CA1, CA3, CTX) (line 543-544), but increased signal and perhaps nuclear distribution in AD (line 397).

Response: Thank you for this suggestion. We have expanded in the Discussion (Page 30)

There is no discussion of supplemental figures 4 and 5 in the main text. Please discuss how aging in these WT and tauKO mice affects MSI1 and MSI2; and whether the changes in MSI1 and MSI2 in Figure 4 are specific to the tau expression. As such, there it is unclear that changes in MSI are tau-dependent (line 748).

Response: Thank you! This is important. We have added a brief description of **Supplementary Figure 4** and **Supplementary Figure 5** and corrected it in the main text (Page 18-19).

Line 566 to 568 – Contradicts figure 4 showing increased nuclear signal from MSI1 in 2-month-old P301L mice? Line 532 “In general, we detected in P301L 2-months old mice a nuclear and axonal MSI1 distribution in hippocampi and cortex”.

Response: Thank you. The text has been corrected according to the data in **Figures 3 and 4**.

Please include secondary only controls to confirm specificity of antibodies for IHC (and either a citation or use of knockout tissue to confirm TOMA2 specificity).

Response: We apologize for this. TOMA2 antibody has been developed and characterized (including quality control and specificity) previously from Kaye Lab; this antibody is one of 4 anti-tau oligomer clones developed. Wolozin group has published TOMA2 antibody in P301S mouse model detecting tau oligomers [9], and other groups are using it too [10].

Figure 5 – in 2month old P301L mice, the low magnification image panel A shows no MSI1/TOMA2 PLA signal in DG. Higher mag image panel D shows robust MSI1/TOMA2 PLA signal in DG nuclei. Could the authors explain this apparent mismatch.

Response: Thank you so much for the observation. This has been corrected. The high magnification images from **c to j (Figure 5)** are PLA (tau13/MSI1 and TOMA-2/MSI1) only from 7-months old mice brains, so it is not related to the 2-months old PLA Panel (**a**). We apologize for this accidental mistake; the legend of **Figure 5c-j** was reported for 2-months, which we have now corrected by reporting 7-months-old mice.

Supplemental Figure 6: The text states that the TOMA2/MSI immunofluorescence is in Supplemental figure 6, but it is in Supplemental Figure 7.

Response: Corrected. Thank you.

Line 648 – It appears that removal of tet does lead to cells without nuclear membrane defects. The 24hr on, 48hr off experimental condition leads to much fewer cells. Is it possible that those cells that had nuclear membrane dysregulation are dying, perhaps indicating poor recovery even after removal of tet?

Response: Thank you. We agree. That this could be the case, since. At 48hr Tet-off, we observed less cells in the field represented but we never found significant differences in classical cell viability assays.

Line 670 – “For euchromatin, we used H3K4me3 and H3K9me3 for heterochromatin”. This phrasing is confusing, which marker was used for which chromatin state?

Response: Corrected. Thank you so much.

Lines 698 to 700 – it is hard to follow the meaning of: “Furthermore, TTC implement interaction (after Tet induction) with LaminB1, while MSI1, as well as for the Histones, showed a marked reduction of interaction (Figure 7I).”

Response: Corrected- Thank you.

There is often use of terms that are somewhat unconventional. The authors should more clearly define their use of the following terms:

What is a “slot plot”. Line 458. Used in figures 2F, G and supp figure 3C.

Response: We truly apologize for that. We have added this to **materials and methods** with references [3], [11] . Slot blot, also called filter trap assay is a technique widely used for the detection and quantification of amyloid aggregates formed by diverse proteins (Page 8).

- “LUT”. Used in figures 3H, 7D.

Response: Corrected: Thank you.

- “increment” and “decrement” (throughout the manuscript, examples: line 671, 673, 684, 696 etc)
- o Usually defined at the “quantity of an increase”

Response: We apologize for this. Corrected, thank you.

- Line 696 - “TTC35” – no indication in text as to what this is.

Response: We apologize for this! This has been corrected: TTC35 is a tau antibody that was generated and validated in our laboratory as published before [12]. It recognizes aggregated forms of tau. We have added this information in the Co-Immunoprecipitation section (**material and methods**) Lines 271-272

- “membrane bumping” line 647.

Response: Corrected, thank you!

Reviewer #3 (Remarks to the Author):

The study by Montalbano et al. demonstrates convincingly that tau oligomers can interact and form soluble aggregates with MSI1 and MSI2 proteins in vitro (HEK cells) and in the mature neurons.

Authors suggest that the pathogenic tau forms such aggregates in the neuron nuclei of the AD, ALS and FTD human and mouse brains that leads to the remodeling of chromatin packing and nuclear membrane structure. Co-localization of MSI1 and MSI2 with tau bundles is also observed in iHEK cells or mouse brain, where the expression of the P301L tau mutant was induced. Latter is known to form similar to the pathogenic tau aggregates.

The results show also that MSI1 is accumulating in nuclei of the aged P301L mice (7 month), but not associated to nucleoli and is not homogeneously distributed. MSI2 is low expressed in 2-month-old mice, but increases with age and detected in neuronal progenitor stem cells and mainly cytoplasmic.

After the performing different co-IPs and the mass-spectroscopic analysis of the nuclear and cytoplasmic fraction (see Fig. 7), the authors conclude that folded nuclear tau and MSI1 regulate the nuclear activity and stabilize chromatin distribution. In contrast, toxic tau conformers interact with MSI2 and destabilize the nuclear envelope, nucleoporins and chromatin localization.

They suggest also that the pathogenic tau interaction with MSI1 and MSI2 proteins in the cell nuclei of the mature neurons can impact nucleocytoplasmic transport and cause lamina disfunction.

All these observations are novel and would be highly interesting for the community focused on the neurodegenerative diseases. This work might be of the great interest for the wider field of scientist as well.

Particularly, I appreciate their extensive mass spectroscopic analysis combined with co-IPs and IF that provides valuable information on how different nucleoporins of nuclear pore complexes (NPCs), importins and NLS-cargos (e.g. different histones, laminA, laminB1, emerin, LAPs etc.) are affected by the accumulation of the toxic/pathogenic tau in the cells.

As shown in Fig. 6 F-H or in Fig. S8 D most of the cargo proteins that bare nuclear localization signal (NLS) are significantly reduced in the nucleus after the Tet-induced expression of toxic P301L tau. It might indicate that the toxic tau affects most likely a nucleocytoplasmic import of these NLS-cargos by preventing their nuclear entry through NPC and release into the nucleus that requires RanGTP. As it was shown previously (Eftekharzadeh et. al, 2018, 2019) toxic tau can co-aggregate with some FG-nucleoporins, e.g. with Nup98 or Nup62, and as such impact NPC permeability. On the other hand, toxic tau is also compromising RanGTP/RanGDP gradient (Eftekharzadeh et. al, 2018, 2019) that would also result in the reduced amount of the NLS-cargo in the cell nuclei. Consequently, an import of LaminA, LaminB1 and other lamina binding

proteins into the nucleus might also be impaired and result in the mis-formation of the proper lamina. Nup50 or Nup153 (both bare NLS-signal and are

delivered into the nucleus by importins during NPC assembly with following incorporation into the nuclear basket) are also reduced in the nuclear fraction and significantly elevated in the cytoplasm that might mean that their delivery into the nucleus is also hindered (but not cellular expression level). It would be beneficial for the paper, if these considerations are also taken into account in the discussion part.

Response: Thank you so much for this precise and concise revision. We truly appreciate your comments, suggestion and clear summary. We expanded and corrected the paragraph discussion section (below) to reflect what you pointed out. Thank you!

“These results indicate that toxic tau may affect the nucleocytoplasmic import of NLS-cargos by preventing their nuclear entry through NPC and release into the nucleus that requires RanGTP. As it was shown toxic tau can co-aggregate with some FG-nucleoproteins, e.g. with Nup98 or Nup62, and as such impact NPC permeability. On the other hand, toxic tau is also compromising RanGTP/RanGDP gradient that would also result in the reduced amount of the NLS-cargo in the cell nuclei. Consequently, an import of LaminA, LaminB1 and other lamina binding proteins into the nucleus might also be impaired and result in the mis-formation of the proper lamina. Nup50 or Nup153 (both bare NLS-signal and are delivered into the nucleus by importins during NPC assembly with following incorporation into the nuclear basket) are also reduced in the nuclear fraction and significantly elevated in the cytoplasm that might indicate that their delivery into the nucleus is also hindered (but not cellular expression level).” (Page 31)

I would highly recommend this paper for the publication in Nature Communication after the minor revisions.

Thank you so much!

Minor:

1. Lane 111: I guess here instead ‘...through binding of the nuclear pore complex (NUP), Nup98.’ It should be written: ‘...through binding of the nuclear pore complex (NPC), e.g. to Nup98’.

Response: Corrected as suggested.

2. Lane 128: does it mean here ‘P301L’ tau mutant (instead of ‘P310L’)?

Response: Thank you! Corrected.

3. Lane 376: and Fig. 1 (G and H) aggregates of tau and MSI 1 and MSI 2 are shown respectively. Authors claim that tau and MSI protein aggregates are co-localizing that is not really obvious from these IF figures.

Response: Response: Thank you for this observation! In **Figure 1g** and **h** we did show quantification of colocalization because as noticed by reviewer we did not observe an intense yellow (overlap) signal. Therefore, we have chosen to show differential regions of interest from the same patients in **Figure 2**. As shown in in **Figure 1g, h** we observed intra-subject variability. Since the colocalization quantification (PCC) could underestimate co-localization of Musashi proteins with TOMA-2 therefore we used regions of interest We also agree with the reviewer's comment that even though we have a positive correlation, the two protein markers do not co-localize completely. However, we also cannot exclude the partial overlap as shown by the ROIs positive correlation (**Supplementary Figure 2**). As described in ALS and FTD the TOMA-2 immunoreactivity is very different with numerous puncta in ALS and more compact in FTD, suggesting, as indicated in recent studies, that different diseases have a peculiar tau deposits - strains. Now we used double the number of cases (6 for each group) studied, in **Supplementary Figure 2**.

We have re-phrased it (Page 16).

4. Fig. S3 D and F: should it be 'MSI2' written in green in the IF headings?

Answer: Corrected. Thank you!

5. Fig 5: when comparing panels C and G; D and H; E and F; and F and G, it seems logic to conclude that the difference in PLA staining between upper and lower panels (e.g. in C) is due to the tau-13 and TOMA-2 staining, because second component (MSI1 antibody is present in both panels). Therefore, it is expected that the same difference in staining should be observed in PLA panels (low versus top) in G (same age, same mice model; same brain area) with extra signal that could be observed (or not) for MSI2 in this case instead of MSI1 like in C. But it seems PLA signal is much weaker here. Why? And why it is interpreted as difference in the MSI1 and MSI2 localization? Same question, when comparing panels D and H in this figure.

Response: Thank you for this consideration. In **Figure 4** we added an Immunoblot for MSI1 and MSI2 (mouse brain homogenates). We observed that MSI1 in aged mice was present largely in multimeric forms compared to young animals, but not a big difference in total amount for MSI2, which makes it hard to detect such small differences by using PLA, which may not reveal. It looks like this is not the case for MSI1 for which we could observe changes in both expression levels and cellular localization.

6. Fig. S7: it is shown that tau is enveloping nuclei, but not really present inside of nuclei as much, but MSI1 is mainly nuclear localized in P301L mice brain. Why is it concluded that they are co-localized in the nucleus?

Response: Thank you! We mentioned the overlap, in the nuclei, between MSI1 and tau oligomers. We have modified **Supplementary Figure 7**; arrows were added to indicate the nuclei where this overlap occurs.

7. Lane 627: LaminB1 and histone 3 level is decreased in the 7 months old P301L mice. Could it be connected with the disruption of NCT (nucleocytoplasmic transport), blockage of NPCs upon oligomeric toxic tau accumulation in the aged mice? If yes, this assumption would be then consistent with the mass spect. data shown in Fig. 6. It seems that nuclear import of NLS-type cargos is impaired and LAPs, LaminA and B and some basket NPC proteins like Nup153, Nup50 are reduced in the nuclear fraction and their amount elevated in the cytoplasm. This could be really an indication that their expression level is not much affected, but their nuclear import is partly inhibited by the aggregating tau.

Response: Thank you! We agree that of NPC impairment provides a strong explanation of nuclear components impairment. We also think that tau oligomers, in the nuclei, interact with NPCs affecting their structural integrity and thus impairing their functions. At this stage, we cannot provide sufficient evidence, but we believe that it relates to chromatin remodeling events.

8. Lane 635: might be better to write 'nuclear proteins' instead of 'receptors';

Response: Corrected

9. Lane 638: 'nucleoporins' instead of 'Nucleoporin' in singular.

Response: Corrected.

10. Fig. 7: IP in Fig. 7G shows that toxic tau interacts with histones directly or via MSI2 (if considering Fig. 7 E and G). For stronger conclusion (shown in Fig. 7 J, K) a direct interaction of toxic tau (e.g. P301L mutant tau) with the recombinant MSI1 and MSI2 or with histones (e.g. histone 3) and lamins would be a plus. It would help to clarify whether toxic tau reduces interaction of MSI1 with histones by binding to them in the nucleus (Fig. 7 H) or less histones are detected in the nuclear fraction of this IP only because histones less efficiently delivered into the nucleus after Tet-induced P301L tau expression (due to change in pore permeability, interaction with tau at the nuclear pores etc.).

Response: Thank you for this consideration! We have included (**Supplementary Figure 9c**) a Western blot of TTC35 IP'ed nuclear/cytoplasm fractions probed for LaminB1 and Histone3. Data confirmed the differences observed in the MassSpec profiles. We also agree that more studies are needed to understand and how Histone3 and other Histones are affected by the mutant and pathological tau.

References

- [1] M. Montalbano *et al.*, “Tau oligomers mediate aggregation of RNA-binding proteins Musashi1 and Musashi2 inducing Lamin alteration.,” *Aging Cell*, p. e13035, Sep. 2019.
- [2] R. Kaye *et al.*, “Common structure of soluble amyloid oligomers implies common mechanism of pathogenesis.,” *Science*, vol. 300, no. 5618, pp. 486–489, Apr. 2003.
- [3] U. Sengupta, M. Montalbano, S. McAllen, G. Minuesa, M. Kharas, and R. Kaye, “Formation of Toxic Oligomeric Assemblies of RNA-binding Protein: Musashi in Alzheimer’s disease.,” *Acta Neuropathol. Commun.*, vol. 6, no. 1, p. 113, Oct. 2018.
- [4] B. Winner *et al.*, “In vivo demonstration that alpha-synuclein oligomers are toxic.,” *Proc. Natl. Acad. Sci. U. S. A.*, vol. 108, no. 10, pp. 4194–4199, Mar. 2011.
- [5] I. R. A. Mackenzie *et al.*, “Nomenclature for neuropathologic subtypes of frontotemporal lobar degeneration: consensus recommendations.,” *Acta Neuropathol.*, vol. 117, no. 1, pp. 15–18, Jan. 2009.
- [6] D. W. Sanders *et al.*, “Distinct tau prion strains propagate in cells and mice and define different tauopathies.,” *Neuron*, vol. 82, no. 6, pp. 1271–1288, Jun. 2014.
- [7] S. K. Kaufman *et al.*, “Tau Prion Strains Dictate Patterns of Cell Pathology, Progression Rate, and Regional Vulnerability In Vivo.,” *Neuron*, vol. 92, no. 4, pp. 796–812, Nov. 2016.
- [8] S. Sakakibara, Y. Nakamura, H. Satoh, and H. Okano, “Rna-binding protein Musashi2: developmentally regulated expression in neural precursor cells and subpopulations of neurons in mammalian CNS.,” *J. Neurosci.*, vol. 21, no. 20, pp. 8091–8107, Oct. 2001.
- [9] L. Jiang *et al.*, “TIA1 regulates the generation and response to toxic tau oligomers.,” *Acta Neuropathol.*, vol. 137, no. 2, pp. 259–277, Feb. 2019.
- [10] Z. Ruan *et al.*, “Alzheimer’s disease brain-derived tau-containing extracellular vesicles: Pathobiology and GABAergic neuronal transmission,” *bioRxiv*, p. 2020.03.15.992719, Jan. 2020.
- [11] F. Lo Cascio *et al.*, “Toxic Tau Oligomers Modulated by Novel Curcumin Derivatives.,” *Sci. Rep.*, vol. 9, no. 1, p. 19011, Dec. 2019.
- [12] U. Sengupta *et al.*, “Tau oligomers in cerebrospinal fluid in Alzheimer’s disease.,” *Ann. Clin. Transl. Neurol.*, vol. 4, no. 4, pp. 226–235, Apr. 2017.

REVIEWERS' COMMENTS:

Reviewer #1 (Remarks to the Author):

This is a revised manuscript, and the authors have addressed satisfactorily to the concerns of this reviewer from the previous submission(s). Consequently, the manuscript is now significantly improved for publication.

Reviewer #2 (Remarks to the Author):

The authors have responded well to my comments. They have an interesting story examining the relationship between tau and MSI1,2 that represents a significant advance for the field. I understand that the COVID19 pandemic interfered with research, and find the manuscript to be much improved.

However, there is still one change that must be made. In lines 583-584, the authors state that "In Tau KO mice, we did not observe big change in MSI1, while we observed a marked signal of MSI2 in cortex and hippocampus." The immunoblots demonstrate a significant age-related change in MSI1 in Tau KO mice. This the authors must state somewhere that "Age causes a significant increase in MSI1 (and change in the distribution of MSI2) in the brain." Otherwise the interpretation is misleading.

So, in summary, a good study but please be rigorous in the interpretation! Ben Wolozin

Reviewer #3 (Remarks to the Author):

In the work of Montalbano and co-workers an influence of pathogenic tau on the localization of MSI1 and MSI2 in the AD, ALS and FRD in brains of human and mouse was carefully analysed. They have explicitly demonstrated that pathogenic tau can interact/co-aggregate with MSI1 and MSI2 and affects their localization. It can also disrupt the nucleocytoplasmic transport and as such affect the nuclear proteins localization and functionality as well as chromatin remodelling. Therefore, I think that a new title of the manuscript ("RNA-Binding Proteins Musashi and Tau soluble aggregates initiate nuclear dysfunction") reflects better its content and conclusions.

The authors have also significantly improved their manuscript and have provided convincing answers to the reviewers. Their results are novel and would significantly contribute into understanding of the pathogenic tau and MSI1 and MSI2 role in the neurodegenerative diseases, such as Alzheimer's disease (AD), amyotrophic lateral sclerosis (ALS) and Frontotemporal dementia (FTD).

Therefore, I would strongly recommend this manuscript for the publication in Nature Communication.

Minor:

I would suggest a few minor changes:

Page 31:

In the added paragraph on this page I would recommend to give a reference in the sentence: "On the other hand, toxic tau is also compromising RanGTP/RanGDP gradient that would also result in the reduced amount of the NLS-cargo in the cell nuclei (Eftekharzadeh et. al, 2018)."

I would also propose to rewrite the sentence in page 31 as follows:

"Nup50 or Nup153, which both have NLS-like-sequences and can also interact with nuclear

transport receptors, are also reduced in the nuclear fraction and significantly elevated in the cytoplasm that might indicate that their recruitment during NPCs assembly is hindered in the presence of the pathogenic tau.”

I think this would be more appropriate.

Authors' response to the reviewers

We would like to thank the reviewers for their important comments and suggestions.

Point-by-point response to the reviewers' comments

Reviewers' comments:

Reviewer #1 (Remarks to the Author):

This is a revised manuscript, and the authors have addressed satisfactorily to the concerns of this reviewer from the previous submission(s). Consequently, the manuscript is now significantly improved for publication.

Thank you for your encouragement.

Reviewer #2 (Remarks to the Author):

The authors have responded well to my comments. They have an interesting story examining the relationship between tau and MSI1,2 that represents a significant advance for the field. I understand that the COVID19 pandemic interfered with research, and find the manuscript to be much improved.

However, there is still one change that must be made. In lines 583-584, the authors state that "In Tau KO mice, we did not observe big change in MSI1, while we observed a marked signal of MSI2 in cortex and hippocampus." The immunoblots demonstrate a significant age-related change in MSI1 in Tau KO mice. This the authors must state somewhere that "Age causes a significant increase in MSI1 (and change in the distribution of MSI2) in the brain." Otherwise the interpretation is misleading.

So, in summary, a good study but please be rigorous in the interpretation! Ben Wolozin

Thank you for your encouragement and valuable suggestion.

Response: Thank you, lines 583-584 have been modified including the sentence suggested by the reviewer.

Reviewer #3 (Remarks to the Author):

In the work of Montalbano and co-workers an influence of pathogenic tau on the localization of MSI1 and MSI2 in the AD, ALS and FRD in brains of human and mouse was carefully analysed. They have explicitly demonstrated that pathogenic tau can interact/co-aggregate with MSI1 and MSI2 and affects their localization. It can also disrupt the nucleocytoplasmic transport and as such affect the nuclear proteins localization and functionality as well as chromatin remodelling. Therefore, I think that a new title of the manuscript ("RNA-Binding Proteins Musashi and Tau soluble aggregates initiate nuclear dysfunction") reflects better its content and conclusions.

The authors have also significantly improved their manuscript and have provided convincing answers to the reviewers. Their results are novel and would significantly contribute into understanding of the pathogenic tau and MSI1 and MSI2 role in the neurodegenerative diseases, such as Alzheimer's disease (AD), amyotrophic lateral sclerosis (ALS) and Frontotemporal dementia (FTD).

Therefore, I would strongly recommend this manuscript for the publication in Nature Communication.

Minor:

I would suggest a few minor changes:

Thank you for your encouragement and valuable suggestions.

Page 31:

In the added paragraph on this page I would recommend to give a reference in the sentence: "On the other hand, toxic tau is also compromising RanGTP/RanGDP gradient that would also result in the reduced amount of the NLS-cargo in the cell nuclei (Eftekharzadeh et. al, 2018)."

Response: Thank you, Reference has been included in page 31.

I would also propose to rewrite the sentence in page 31 as follows:

"Nup50 or Nup153, which both have NLS-like-sequences and can also interact with nuclear transport receptors, are also reduced in the nuclear fraction and significantly elevated in the cytoplasm that might indicate that their recruitment during NPCs assembly is hindered in the presence of the pathogenic tau."

I think this would be more appropriate.

Response: Thank you, the sentence has been rewritten following reviewer suggestion in page 31.